# PROVABLE BENEFITS OF MULTI-TASK RL UNDER NON-MARKOVIAN DECISION MAKING PROCESSES

**Ruiquan Huang**[*†]          **Yuan Cheng**[*‡]          **Jing Yang**[†]

**Vincent Tan**[‡]                          **Yingbin Liang**[§]

## ABSTRACT

In multi-task reinforcement learning (RL) under Markov decision processes (MDPs), the presence of shared latent structures among multiple MDPs has been shown to yield significant benefits to the sample efficiency compared to single-task RL. In this paper, we investigate whether such a benefit can extend to more general sequential decision making problems such as predictive state representations (PSRs). The main challenge here is that the large and complex model space makes it hard to identify what types of common latent structure of multi-task PSRs can reduce the model complexity and improve sample efficiency. To this end, we posit a *joint model class* for tasks and use the notion of $\eta$-bracketing number to quantify its complexity; this number also serves as a general metric to capture the similarity of tasks and thus determines the benefit of multi-task over single-task RL. We first study upstream multi-task learning over PSRs, in which all tasks share the same observation and action spaces. We propose a provably efficient algorithm UMT-PSR for finding near-optimal policies for all PSRs, and demonstrate that the advantage of multi-task learning manifests if the joint model class of PSRs has a smaller $\eta$-bracketing number compared to that of individual single-task learning. We further investigate downstream learning, in which the agent needs to learn a new target task that shares some commonalities with the upstream tasks via a similarity constraint. By exploiting the learned PSRs from the upstream, we develop a sample-efficient algorithm that provably finds a near-optimal policy. Upon specialization to some examples with small $\eta$-bracketing numbers, our results further highlight the benefit compared to directly learning a single-task PSR.

## 1 INTRODUCTION

Multi-task sequential decision making, or multi-task reinforcement learning (MTRL) is a subfield of reinforcement learning (RL) that extends the learning process across multiple tasks. Many real-world applications can be modeled by MTRL. For instance, in robotics and autonomous driving, different types of robots and vehicles in a shared environment can have different observational capabilities based on their sensors and learning goals. Other applications include personalized healthcare, weather forecasting across different regions, and manufacturing quality control on different types of products. The fundamental idea behind MTRL is to leverage the inherent similarities among a set of tasks in order to improve the overall learning efficiency and performance. For Markov decision processes (MDPs), a line of works (Pathak et al., 2017; Tang et al., 2017; Oord et al., 2018; Laskin et al., 2020; Lu et al., 2021; Cheng et al., 2022; Agarwal et al., 2022; Pacchiano et al., 2022) have explored multi-task representation learning and shown its benefit both practically and theoretically.

However, it is still an open question whether such a benefit can extend to more general sequential decision making problems, even in partially observable MDPs (POMDPs), let alone more general predictive state representations (PSRs). In this context, it is even unclear:

---

[*]Equal contribution.

[†]Penn State University, State College, PA 16801, USA. `{rzh514,yangjing}@psu.edu`

[‡]National University of Singapore, 119077, Singapore.  `yuan.cheng@u.nus.edu, vtan@nus.edu.sg`

[§]Ohio State University, Columbus, OH 43210, USA. `liang.889@osu.edu`

*When can latent similarity structure encompassed by multiple PSRs be potentially beneficial?*

The challenges mainly emanate from two aspects. First, *the large and complex model space makes it hard to identify what types of common latent structure of multi-task PSRs can reduce the model complexity*. The non-Markovian property of these problems implies that the sufficient statistics or belief about the current environmental state encompasses all the observations and actions from past interactions with the environment. This dramatically increases the statistical complexity. Even for a finite observation space and action space, model complexity can be exponentially large in the number of observations and actions. Such a complex parameter space makes it difficult to identify what types of latent similarity structure of multi-task PSRs reduce the model complexity. Second, *reduced* **model complexity** *does not necessarily result in benefit in* **statistical efficiency** *gain of RL*. In RL, model learning and data collection are intertwined. The agent has to choose an exploration policy in each iteration based on the model learned in the past. Such iterative process introduces temporal dependence to the collected data, which makes the analysis of multi-task PSRs complicated.

In this paper, we answer the question above with upstream multi-task learning and downstream transfer learning. We summarize our contributions below.

1. To deal with the first challenge, we propose a unified approach to characterize the effect of task similarity on model complexity by introducing the notion of the $\eta$-*bracketing number* for the *joint model space* of multiple tasks. Regardless of whether the concrete form of task similarity is implicit or explicit, desirable task similarity should contribute to reduce the $\eta$-bracketing number compared to that without similarity structures. This significantly generalizes existing studies of multi-task MDPs that considered only specific task similarity structures.

2. We deal with the second challenge in both upstream and downstream learning. For the former, we propose a novel multi-task PSRs algorithm called UMT-PSR, which features a pairwise additive distance-based optimistic planning and exploration as well as confidence set construction based on the bracketing number of the *joint* model class. We then prove that if the bracketing number of the multi-task model class normalized by the number of tasks is lower than that of a single task, UMT-PSR benefits from multi-task learning with these novel designs. We then provide several specific multi-task POMDP/PSR examples with low bracketing number to demonstrate that UMT-PSR is often more efficient than single-task learning.

3. We further employ the upstream learning to downstream learning by connecting upstream and downstream models via similarity constraints. We show that the downstream learning can identify a near-accurate model and find a near-optimal policy. Upon specialization to the examples used to elucidate the $\eta$-bracketing numbers, our downstream results further highlight the benefit in comparison to directly learning parameters of PSRs without upstream information. Our analysis here features a novel technique of using Rényi divergence to measure the approximation error which guarantees the sub-optimality bound without requiring the realizability condition.

Our work is the first theoretical study that characterizes the benefits of multi-task RL with PSRs/POMDPs over its single-task counterpart.

## 2 RELATED WORK

**MTRL under MDPs:** Multitask representation learning and transfer learning have been extensively studied in RL, particularly under MDPs. Arora et al. (2020) demonstrated that representation learning can reduce sample complexity for imitation learning. Hu et al. (2021) analyzed MTRL with low inherent Bellman error (Zanette et al., 2020) and known representation. Zhang & Wang (2021) studied multi-task learning under similar transition kernels. In contrast, Brunskill & Li (2013) studied the benefit of MTRL when each task is independently sampled from a distribution over a finite set of MDPs. Recent studies have also considered the case where all tasks share a common representation, including D'Eramo et al. (2020) which demonstrated the convergence rate benefit on value iteration, and Lu et al. (2021) which proved the sample efficiency gain of MTRL under low-rank MDPs. Some recent work further took the impact of sequential exploration and temporal dependence in data into account. Considering sequential exploration with shared unknown representation, Cheng et al. (2022); Agarwal et al. (2022) studied reward free MTRL under low-rank MDPs as upstream learning and applied the learned representation from upstream to downstream RL. Pacchiano et al. (2022) focused on a common low-dimensional linear representation and investigated MTRL under linearly-factored

MDPs. Lu et al. (2022) explored MTRL with general function approximation. There also exist a few papers studying multi-task POMDPs from theoretical (Li et al., 2009) and practical (Omidshafiei et al., 2017) side. Note that all the above studies considered specific common model structures shared among tasks, whereas our paper proposes a unified way to characterize the similarity among tasks. Further, none of the existing studies considered model-based multi-task PSRs, which is the focus of our paper.

**Single-task RL with PSRs and general sequential decision making problems:** A general decision making framework PSR (Littman & Sutton, 2001) was proposed to generalize MDPs and POMDPs. Since then, various approaches have been studied to make the problem tractable with polynomial sample efficiency. These methods include spectral type of techniques (Boots et al., 2011; Hefny et al., 2015; Jiang et al., 2018; Zhang et al., 2022), methods based on optimistic planning and maximum log-likelihood estimators together with confidence set-based design (Zhan et al., 2022; Liu et al., 2022), the bonus-based approaches (Huang et al., 2023), value-based actor-critic approaches (Uehara et al., 2022), posterior sampling methods (Zhong et al., 2022). Chen et al. (2022) further improved the sample efficiency for previous work including OMLE (Liu et al., 2022), MOPS (Agarwal & Zhang, 2022), and E2D (Foster et al., 2021).

## 3 PRELIMINARIES

**Notations.** For any positive integer $N$, we use $[N]$ to denote the set $\{1, \cdots, N\}$. For any vector $x$, the $i$-th coordinate of $x$ is represented as $[x]_i$. For a set $\mathcal{X}$, the Cartesian product of $N$ copies of $\mathcal{X}$ is denoted by $\mathcal{X}^N$. For probability distributions $\mathbb{P}$ and $\mathbb{Q}$ supported on a countable set $\mathcal{X}$, the total variation distance between them is $\mathsf{D}_{\mathsf{TV}}\left(\mathbb{P}, \mathbb{Q}\right) = \sum_x |\mathbb{P}(x) - \mathbb{Q}(x)|$, and the Rényi divergence of order $\alpha$, for $\alpha > 1$, between them is $\mathsf{D}_{\mathsf{R},\alpha}(\mathbb{P}, \mathbb{Q}) = \frac{1}{\alpha-1} \log \mathbb{E}_{\mathbb{P}}[(d\mathbb{P}/d\mathbb{Q})^{\alpha-1}]$.

### 3.1 THE NON-MARKOVIAN DECISION MAKING PROBLEM

We consider an episodic decision making process, which is generally non-Markovian, with an observation space $\mathcal{O}$ and a finite action space $\mathcal{A}$. We assume that the process is episodic and each episode contains $H$ steps, i.e., with horizon $H$. At each step, the evolution of the process is controlled by an underlying distribution $\mathbb{P}$, where $\mathbb{P}(o_h|o_1, \ldots, o_{h-1}, a_1, \ldots, a_{h-1})$ is the probability of visiting $o_h$ at step $h$ given that the learning agent has observed $o_t \in \mathcal{O}$ and taken action $a_t \in \mathcal{A}$ for previous steps $t \in [h-1]$. And the learning agent receives a reward at each episode determined by the reward function $R : (\mathcal{O} \times \mathcal{A})^H \to [0, 1]$. We denote such a process compactly as $\mathsf{P} = (\mathcal{O}, \mathcal{A}, H, \mathbb{P}, R)$. For each step $h$, we denote historical trajectory as $\tau_h := (o_1, a_1, \ldots, o_h, a_h)$, the set of all possible historical trajectories as $\mathcal{H}_h = (\mathcal{O} \times \mathcal{A})^h$, the future trajectory as $\omega_h := (o_{h+1}, a_{h+1}, \ldots, o_H, a_H)$, and the set of all possible future trajectories as $\Omega_h = (\mathcal{O} \times \mathcal{A})^{H-h}$.

The agent interacts with the environment in each episode as follows. At step 1, a fixed initial observation $o_1$ is drawn. At each step $h \in [H]$, due to the non-Markovian nature, the action selection and environment transitions are based on whole history information. Specifically, the agent can choose an action $a_h$ based on the history $\tau_{h-1}$ and the current observation $o_h$ with a strategy (probability) $\pi_h(a_h|\tau_{h-1}, o_h)$. We denote such a strategy as a policy, and collect the policies over $H$ steps into $\pi = \{\pi_h\}_{h=1}^H$, and denote the set of all feasible policies as $\Pi$. Then the environment takes a transition to $o_{h+1}$ based on $\mathbb{P}(o_{h+1}|\tau_h)$. The episode terminates after $H$ steps.

For any historical trajectory $\tau_h$, we further divided it into $\tau_h^o = (o_1, \ldots, o_h)$ and $\tau_h^a = (a_1, \ldots, a_h)$ which is observation and action sequences contained in $\tau_h$, respectively. Similar to $\tau_h$, for the future trajectories $\omega_h$, we denote $\omega_h^o$ as the observation sequence in $\omega_h$, and $\omega_h^a$ as the action sequence in $\omega_h$. For simplicity, we write $\pi(\tau_h) = \pi(a_h|o_h, \tau_{h-1}) \cdots \pi(a_1|o_1)$ to denote the probability of choosing the sequence of actions $\tau_h^a$ given the observations $\tau_h^o$ under the policy $\pi$. We denote $\mathbb{P}^\pi$ as the distribution of the trajectories induced by the policy $\pi$ under the dynamics $\mathbb{P}$. The value function of a policy $\pi$ under $\mathbb{P}$ and the reward $R$ is denoted by $V_{\mathbb{P},R}^\pi = \mathbb{E}_{\tau_H \sim \mathbb{P}^\pi}[R(\tau_H)]$. The primary learning goal is to find an $\epsilon$-optimal policy $\bar{\pi}$, which is one that satisfies $\max_\pi V_{\mathbb{P},R}^\pi - V_{\mathbb{P},R}^{\bar{\pi}} \leq \epsilon$.

Given that addressing a general decision-making problem entails an exponentially large sample complexity in the worst case, this paper focuses on the *low-rank* class of problems as in Zhan et al. (2022); Liu et al. (2022); Chen et al. (2022). Before formal definition of the low-rank problem, we

introduce the dynamics matrix $\mathbb{D}_h \in \mathbb{R}^{|\mathcal{H}_h| \times |\Omega_h|}$ for each $h$, where we use $\tau_h \in \mathcal{H}_h$ and $\omega_h \in \Omega_h$ to index the rows and columns of the matrix $\mathbb{D}_h$, respectively, and the entry at the $\tau_h$-th row and $\omega_h$-th column of $\mathbb{D}_h$ equals to the conditional probability $\mathbb{P}(\omega_h^o, \tau_h^o | \tau_h^a, \omega_h^a)$.

**Definition 1** (Rank-$r$ sequential decision making problem). *A sequential decision making problem is rank $r$ if for any $h$, the model dynamics matrix $\mathbb{D}_h$ has rank at most $r$.*

As a result, for each $h$, the probability of observing $\omega_h^o$ can be represented by a linear combination of probabilities on a set of future trajectories known to the agent called *core tests* $\mathcal{Q}_h = \{\mathbf{q}_h^1, \ldots, \mathbf{q}_h^{d_h}\} \subset \Omega_h$, where $d_h \geq r$. Specifically, there exist functions $\mathbf{m} : \Omega_h \to \mathbb{R}^{d_h}, \psi : \mathcal{H}_h \to \mathbb{R}^{d_h}$ such that $(i)$ the value of the $\ell$-th coordinate of $\psi(\tau_h)$ equals to the conditional probability $\mathbb{P}(\mathbf{o}_h^\ell, \tau_h^o | \mathbf{a}_h^\ell, \tau_h^a)$ on $(\mathbf{q}_h^\ell, \tau_h)$, where $\mathbf{o}_h^\ell$ and $\mathbf{a}_h^\ell$ to denote the observation and the action sequences of $\mathbf{q}_h^\ell$, and $(ii)$ for any $\omega_h \in \Omega_h, \tau_h \in \mathcal{H}_h$, the conditional probability can be factorized as

$$\mathbb{P}(\omega_h^o, \tau_h^o | \tau_h^a, \omega_h^a) = \mathbf{m}(\omega_h)^\top \psi(\tau_h). \tag{1}$$

**Predictive State Representation.** Following from Theorem C.1 in Liu et al. (2022), given core tests $\{\mathcal{Q}_h\}_{h=1}^H$, any low rank decision making problem admits a (self-consistent) **predictive state representation** (PSR) $\theta = \{(\phi_h, \mathbf{M}_h)\}_{h=1}^H$, such that Eq. 1 can be reparameterized by $\theta$. Mathematically, For any $h \in [H], \tau_h \in \mathcal{H}_h, \omega_h \in \Omega_h$:

$$\mathbf{m}(\omega_h)^\top = \phi_H^\top \mathbf{M}_H(o_H, a_H) \cdots \mathbf{M}_{h+1}(o_{h+1}, a_{h+1}), \quad \psi(\tau_h) = \mathbf{M}_h(o_h, a_h) \cdots \mathbf{M}_1(o_1, a_1) \psi_0,$$

and $\sum_{(o_h, a_h) \in \mathcal{O} \times \mathcal{A}} \phi_{h+1}^\top \mathbf{M}_h(o_h, a_h) = \phi_h^\top$. For ease of the presentation, we assume $\psi_0$ is known.[1]

The following assumption is standard in the literature (Liu et al., 2022; Chen et al., 2022).

**Assumption 1** ($\gamma$-well-conditioned PSR). *We assume any PSR $\theta = \{(\phi_h, \mathbf{M}_h)\}_{h=1}^H$ considered in this paper is $\gamma$-well-conditioned for some $\gamma > 0$, i.e.*

$$\forall h \in [H], \max_{x \in \mathbb{R}^{d_h} : \|x\|_1 \leq 1} \max_{\pi \in \Pi} \max_{\tau_h \in \mathcal{H}_h} \sum_{\omega_h \in \Omega_h} \pi(\omega_h | \tau_h) |\mathbf{m}(\omega_h)^\top x| \leq \frac{1}{\gamma}. \tag{2}$$

In the following context, we use $\mathbb{P}_\theta$ to indicate the model determined by the PSR $\theta$. For simplicity, we denote $V_{\mathbb{P}_\theta, R}^\pi$ as $V_{\theta, R}^\pi$. Moreover, let $\mathcal{Q}_h^A = \{\mathbf{a}_h^\ell\}_{\ell=1}^{d_h}$ be the action sequence set from core tests which is constructed by eliminating any repeated action sequence. The set $\mathcal{Q}_h^A$ is also known as the core action sequence set. The set of all rank-$r$ and $\gamma$-well-conditioned PSRs is denoted by $\Theta$.

## 3.2 UPSTREAM MULTI-TASK LEARNING

In **upstream multi-task** learning, the agent needs to solve $N$ low-rank decision making problems (also known as source tasks) at the same time instead of only one single problem (task). The set of $N$ source tasks is denoted by $\{\mathsf{P}_n\}_{n=1}^N$, where $\mathsf{P}_n = (\mathcal{O}, \mathcal{A}, H, \mathbb{P}_{\theta_n^*}, R_n)$, and $\theta_n^* \in \Theta$.[2] In other words, all $N$ tasks are identical except for their model parameters $\theta_n^* = \{(\phi_h^{n,*}, \mathbf{M}_h^{n,*})\}_{h=1}^H$, and reward functions $R_n$. Moreover, we denote the model class of multi-task PSRs as $\boldsymbol{\Theta}_\mathrm{u}$ (the subscript stands for upstream), a subset of $\Theta^N$.

The goal of the upstream learning consists of two parts: (i) Finding near-optimal policies for all $N$ tasks on average. Mathematically, given an accuracy level $\epsilon$, the set of $N$ policies that are produced by the algorithm $\{\bar{\pi}^1, \ldots, \bar{\pi}^N\}$ should satisfy $\frac{1}{N} \sum_{n=1}^N (\max_\pi V_{\theta_n^*, R_n}^\pi - V_{\theta_n^*, R_n}^{\bar{\pi}^n}) \leq \epsilon$; (ii) Characterizing the theoretical benefit of multi-task PSRs learning in terms of the sample complexity, compared to learning each task individually.

## 3.3 BRACKETING NUMBER OF JOINT PARAMETER SPACE

One critical factor that affects the efficiency of multi-task learning compared to separate task learning is the presence of shared latent structure among the tasks, which yields a reduced model space in

---

[1]The sample complexity of learning $\psi_0$ if it is unknown is relatively small compared to the learning of the other parameters (Liu et al., 2022).

[2]For simplicity, we assume all tasks have the same rank and $\gamma$, but have different core test sets. The extension to different ranks and $\gamma$'s is straightforward.

multi-task PSRs learning, as compared to separately learning single tasks over the Cartesian product of $N$ model spaces (see Figure 1 for an illustration in 2 dimensions). Consequently, this reduction in model complexity can ultimately lead to improved sample efficiency. Unlike the specific shared model structures among multiple tasks that the previous works studied, such as shared representation in Cheng et al. (2022) and similar transition kernels in Zhang & Wang (2021), here we focus on a general shared model space and use the notion of the $\eta$-**bracketing number** to quantify the complexity of the joint model space. Such a notion plays a central role in capturing the benefit of multi-task PSR learning over single-task learning.

We start with a domain $\mathcal{X}$ and a single task function class $\mathcal{F}$, in which each element $f : \mathcal{X} \to \mathbb{R}_+$. For the multi-task case, the function class is a subset $\mathcal{F}_u$ of $\mathcal{F}^N$.

**Definition 2** ($\eta$-Bracketing number of vector-valued function class $\mathcal{F}_u$ w.r.t. $\|\cdot\|$). *Given two vector-valued functions* $\mathbf{l}$ *and* $\mathbf{g} : \mathcal{X} \to \mathbb{R}_+^N$, *the bracket* $[\mathbf{l}, \mathbf{g}]$ *is the set of all functions* $\mathbf{f} \in \mathcal{F}_u$ *satisfying* $\mathbf{l} \le \mathbf{f} \le \mathbf{g}$.[3] *An* $\eta$-*bracket is a bracket* $[\mathbf{l}, \mathbf{g}]$ *with* $\|\mathbf{g} - \mathbf{l}\| < \eta$. *The bracketing number* $\mathcal{N}_\eta(\mathcal{F}_u, \|\cdot\|)$ *is the minimum number of* $\eta$-*brackets needed to cover* $\mathcal{F}_u$.[4]

In this paper, we are interested in the bracketing number of the *joint* model space, i.e., distribution spaces over $(\mathcal{O} \times \mathcal{A})^H$ parameterized by $\boldsymbol{\Theta}_u$. For simplicity, we use $\mathcal{N}_\eta(\boldsymbol{\Theta}_u)$ to denote the $\eta$-bracketing number of $\{(\mathbb{P}_{\theta_1}, \ldots, \mathbb{P}_{\theta_N}) | \boldsymbol{\theta} \in \boldsymbol{\Theta}_u\}$ w.r.t. the $\ell_\infty$ policy weighted norm $\|\cdot\|_\infty^P$, where the $\ell_\infty$ policy weighted norm between two vector-valued functions $\mathbf{l} = \{l_1, \ldots, l_N\}$ and $\mathbf{g} = \{g_1, \ldots, g_N\}$ defined on $(\mathcal{O} \times \mathcal{A})^H$ is equal to $\|\mathbf{g} - \mathbf{l}\|_\infty^P = \max_{i \in [N]} \max_{\pi_i \in \Pi} \sum_{\tau_H} |l_i(\tau_H) - g_i(\tau_H)| \pi_i(\tau_H)$. As we will show later, a lower $\eta$-bracketing number of the joint model space results in a lower sample complexity in multi-task PSR learning.

In practice, it is common tasks share certain common model structures and hence their joint model space will have a much lower $\eta$-bracketing number compared to the product of model spaces (i.e, treating the model of each task separately). We provide several such examples of non-Markovian decision processes in Section 4.3. We provide more examples of MDPs with their $\eta$-bracketing numbers in Appx. F. Notably, there can be much richer scenarios beyond these examples.

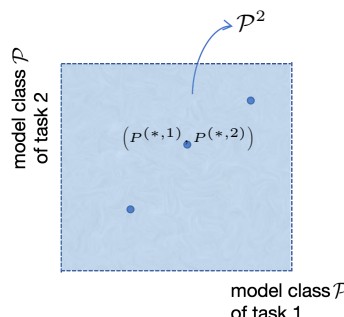

(a) Independent learning of each task where the joint model class is $\mathcal{P}^2$

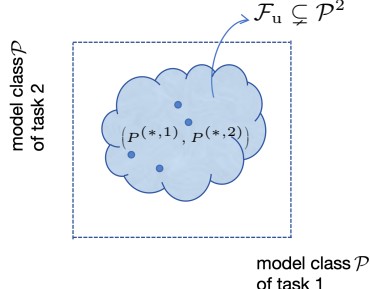

(b) Joint learning of tasks with shared latent model structure. The joint model class is $\mathcal{F}_u$, a strict subset of $\mathcal{P}^2$.

Figure 1: Reduction in $\eta$-bracketing number when $\mathcal{F}_u \subsetneq \mathcal{P}^2$

### 3.4 DOWNSTREAM TRANSFER LEARNING

In **downstream learning**, the agent is assigned with a new target task $\mathsf{P}_0 = (\mathcal{O}, \mathcal{A}, H, \mathbb{P}_{\theta_0^*}, R_0)$, where $\theta_0^* \in \Theta$, which shares some similarities with source tasks to benefit from upstream learning. Here, we capture the shared structure between upstream and downstream tasks via the **similarity constraint** $\mathsf{C}(\theta_0, \theta_1^*, \ldots \theta_N^*) \le \mathbf{0}$ where $\mathsf{C} : \Theta^{N+1} \to \mathbb{R}^{n_d}$, $n_d \in \mathbb{N}$. The similarity constraint establishes the relationship between the downstream target task and the upstream source tasks. Hence, the downstream model class is given by $\Theta_0^u = \{\theta_0 \in \Theta | \mathsf{C}(\theta_0, \theta_1^*, \ldots \theta_N^*) \le \mathbf{0}\}$. We note that the similarity constraint is general enough to capture various relationships between upstream and downstream tasks. For example, Cheng et al. (2022) consider the case when the downstream task shares the same representation as upstream tasks, which is equivalent to assuming $[\mathsf{C}(\theta_0, \ldots, \theta_N^*)]_n = \|\phi^{(*,n)} - \phi_0\|_2$, where $n \in [N]$ and $\phi^{(*,n)}$ is the representation of task $n$. However, the similarity constraint allows much richer beyond the above example, for example, downstream tasks can have similar but not the same representations as the upstream, or may share only some representation features, but not all of them.

---

[3]We write that two vectors $\mathbf{a}, \mathbf{b}$ satisfy $\mathbf{a} \le \mathbf{b}$ if $\mathbf{b} - \mathbf{a}$ is coordinate-wise nonnegative.
[4]We say that a collection of sets $S_1 \ldots, S_n$ *cover* a set $S$ if $S \subset \cup_{i=1}^n S_i$.

The goal of the downstream learning is to find a near-optimal policy, by exploiting the constraint similarity with upstream tasks and utilizing upstream knowledge to achieve better sample efficiency compared with learning without upstream knowledge.

# 4 UPSTREAM LEARNING OVER MULTI-TASK PSRS

We present our upstream algorithm in Section 4.1, characterize its theoretical performance in Section 4.2 and present examples to validate the benefit of upstream multi-task learning in Section 4.3.

We use bold symbol to represent the multi-task parameters or policy. Specifically, $\boldsymbol{\theta} = (\theta_1, \ldots, \theta_N)$, and $\boldsymbol{\pi} = (\pi_1, \ldots, \pi_N)$. We define $Q_A = \max_n \max_h |\mathcal{Q}_h^{n,A}|$, where $Q_h^{n,A}$ is the core action sequence set of task $n$ at step $h$. The policy, denoted by $\nu_h(\pi, \pi')$, takes $\pi$ at the initial $h-1$ steps and switches to $\pi'$ from the $h$-th step. Lastly, $\mathbf{u}_{\mathcal{X}}$ represents the uniform distribution over the set $\mathcal{X}$.

## 4.1 ALGORITHM: UPSTREAM MULTI-TASK PSRS (UMT-PSR)

We provide the pseudo-code of our upstream multi-task algorithm called *Upstream Multi-Task PSRs* (UMT-PSR) in Algorithm 1. This iterative algorithm consists of three main steps as follows.

---

**Algorithm 1** Upstream **M**ulti-**T**ask **PSR**s (UMT-PSR)

1: **Input:** $\mathcal{B}_1 = \boldsymbol{\Theta}$ model class, estimation margin $\beta^{(N)}$, maximum iteration number $K$.
2: **for** $k = 1, \ldots, K$ **do**
3:     Set $\boldsymbol{\pi}^k = \arg\max_{\boldsymbol{\pi} \in \Pi^N} \max_{\boldsymbol{\theta}, \boldsymbol{\theta}' \in \mathcal{B}_k} \sum_{n \in [N]} \mathtt{D}_{\mathsf{TV}}\left(\mathbb{P}_{\theta_n}^{\pi^n}, \mathbb{P}_{\theta_n'}^{\pi^n}\right)$
4:     **for** $n, h \in [N] \times [H]$ **do**
5:         Use $\nu_h^{\pi^{n,k}}$ to collect data $\tau_H^{n,k,h}$.
6:     **end for**
7:     Construct $\mathcal{B}_{k+1} =$

$$
\left\{ \boldsymbol{\theta} \in \boldsymbol{\Theta}_{\mathbf{u}} : \sum_{\substack{t \in [k], h \in [H] \\ n \in [N]}} \log \mathbb{P}_{\theta_n}^{\nu_h^{\pi^{n,t}}}(\tau_H^{n,t,h}) \geq \max_{\boldsymbol{\theta}' \in \boldsymbol{\Theta}_{\mathbf{u}}} \sum_{\substack{t \in [k], h \in [H] \\ n \in [N]}} \log \mathbb{P}_{\theta_n'}^{\nu_h^{\pi^{n,t}}}(\tau_H^{n,t,h}) - \beta^{(N)} \right\} \cap \mathcal{B}_k.
$$

8: **end for**
9: **Output:** Any $\overline{\boldsymbol{\theta}} \in \mathcal{B}_{K+1}$, and a greedy multi-task policy $\overline{\boldsymbol{\pi}} = \arg\max_{\boldsymbol{\pi}} \sum_{n \in [N]} V_{\overline{\theta}_n, R_n}^{\pi^n}$

---

**Pairwise additive distance based multi-task planning** (Line 3): To promote joint planning among tasks, a natural choice to measure the distance between two multi-task models is the distance between the two product distributions $\mathbb{P}_{\theta_1}^{\pi_1} \times \cdots \times \mathbb{P}_{\theta_N}^{\pi_N}$ and $\mathbb{P}_{\theta_1'}^{\pi_1} \times \cdots \times \mathbb{P}_{\theta_N'}^{\pi_N}$. However, such a "distance between product distributions" is not sufficient to guarantee the accuracy of the individual models of each task, which is needed in the analysis of the sum of the individual value functions. Hence, we propose to use the "pairwise additive distance" for our planning, defined as $\mathtt{D}_{\boldsymbol{\pi}}(\boldsymbol{\theta}, \boldsymbol{\theta}') \triangleq \sum_{n \in [N]} \mathtt{D}_{\mathsf{TV}}(\mathbb{P}_{\theta_n}^{\pi^n}, \mathbb{P}_{\theta_n'}^{\pi^n})$.

More specifically, at each iteration $k$, UMT-PSR selects a multi-task policy $\boldsymbol{\pi}^k = (\pi^{1,k}, \ldots, \pi^{N,k})$ that maximizes the largest pairwise additive distance $\max_{\boldsymbol{\theta}, \boldsymbol{\theta}'} \mathtt{D}_{\boldsymbol{\pi}}(\boldsymbol{\theta}, \boldsymbol{\theta}')$ within the confidence set $\mathcal{B}_k$ (which will be specified later). An important property of $\mathcal{B}_k$ is that it contains the true model $\boldsymbol{\theta}^*$ with high probability. Using this property, the largest pairwise additive distance serves as an *optimistic* value of the uncertainty $\mathtt{D}_{\boldsymbol{\pi}}(\boldsymbol{\theta}^*, \boldsymbol{\theta})$ for any multi-task model $\boldsymbol{\theta} \in \mathcal{B}_k$.

**Multi-task exploration** (Line 5): Building upon the planning policy $\boldsymbol{\pi}^k$, for each task $n$ and each step $h$, UMT-PSR executes the policy $\pi^{n,k}$ for first $h-1$ steps, and then uniformly selects an action sequence in $\mathcal{A} \times \mathcal{Q}_h^{n,A}$ for the following $H - h + 1$ steps. In particular, at step $h$, UMT-PSR uniformly takes an action in $\mathcal{A}$, and then uniformly chooses a core action sequence $\mathbf{a}_h$ such that regardless of what the observation sequence is, UMT-PSR always plays the action in the sampled core action sequence. In summary, for each $(n, h) \in [N] \times [H]$, UMT-PSR adopts the policy $\nu_h(\pi^{n,k}, \mathbf{u}_{\mathcal{A} \times \mathcal{Q}_h^{n,A}})$ to collect a sample trajectory $\tau_H^{n,k,h}$. We abbreviate $\nu_h(\pi^{n,k}, \mathbf{u}_{\mathcal{A} \times \mathcal{Q}_h^{n,A}})$ as $\nu_h^{\pi^{n,k}}$.

**Confidence set construction via bracketing number of joint model class** (Line 7): Given the sampled trajectories, UMT-PSR calls a maximum likelihood estimation oracle to construct the multi-task confidence set. A novel element here is the use of the bracketing number of the *joint model class* to characterize estimation margin $\beta^{(N)}$, which is an upper bound of the gap between the maximum log-likelihood within $\Theta_u$ and the log-likelihood of the true model. Such a design provides a unified way for any MTRL problem and avoids individual design for each problem in a case-by-case manner.

## 4.2 MAIN THEORETICAL RESULT

The following theorem characterizes the guarantee of the model estimation and the sample complexity to find a near-optimal multi-task policy.

**Theorem 1.** *Under Assumption 1 , for any fixed $\delta > 0$, let $\Theta_u$ be the multi-task parameter space, $\beta^{(N)} = c_1(\log \frac{KHN}{\delta} + \log \mathcal{N}_\eta(\Theta_u))$, where $c_1 > 0$ and $\eta \leq \frac{1}{KHN}$. Then with probability at least $1 - \delta$, UMT-PSR finds a multi-task model $\overline{\boldsymbol{\theta}} = (\bar{\theta}_1, \ldots, \bar{\theta}_N)$ such that*

$$\sum_{n=1}^{N} \max_{\pi^n \in \Pi} \mathrm{D_{TV}}\left(\mathbb{P}_{\bar{\theta}_n}^{\pi^n}, \mathbb{P}_{\theta_n^*}^{\pi^n}\right) \leq \tilde{O}\left(\frac{Q_A}{\gamma}\sqrt{\frac{rH|\mathcal{A}|N\beta^{(N)}}{K}}\right).$$

*In addition, if $K = \frac{c_2 r|\mathcal{A}|Q_A^2 H\beta^{(N)}}{N\gamma^2\epsilon^2}$ for large enough $c_2 > 0$, UMT-PSR produces a multi-task policy $\overline{\boldsymbol{\pi}} = (\bar{\pi}^1, \ldots, \bar{\pi}^N)$ such that the average sub-optimality gap is at most $\epsilon$, i.e.*

$$\frac{1}{N}\sum_{n=1}^{N}\left(\max_{\pi \in \Pi} V_{\theta_n^*, R_n}^{\pi} - V_{\theta^*, R_n}^{\bar{\pi}^n}\right) \leq \epsilon. \tag{3}$$

**Benefits of multi-task learning:** Theorem 1 shows that with the sample complexity $\tilde{O}(\frac{r|\mathcal{A}|Q_A^2 H^2\beta^{(N)}}{N\gamma^2\epsilon^2})$, UMT-PSR identifies an $\epsilon$-optimal multi-task policy. As a comparison, the best known sample complexity of a single-task PSR RL is given by $O(\frac{r|\mathcal{A}|Q_A^2 H^2\beta}{\gamma^2\epsilon^2})$ in Chen et al. (2022), where $\beta^{(1)} = \tilde{O}(r^2|\mathcal{O}||\mathcal{A}|H^2)$ scales the logarithm of the bracketing number of a single-task PSR with rank $r$. It is clear that as long as $\beta^{(N)} < N\beta^{(1)}$, then UMT-PSR enjoys multi-task benefit in the sample complexity. In Section 4.3, we will provide several example multi-task POMDPs/PSRs to illustrate that such a condition can be satisfied broadly.

Next, we make a few comparisons concerning $\beta^{(N)}$. **(i)** If $N = 1$, Theorem 1 matches the best known sample complexity given in Chen et al. (2022). **(ii)** If none of tasks share any similarity, i.e., $\Theta_u = \Theta^N$, we have $\beta^{(N)} = N\beta^{(1)}$, and the sample complexity does not exhibit any benefit compared to learning the tasks separately. This coincides with the intuition that in the worst case, multi-task learning is not required. **(iii)** The benefits of multi-task learning are more evident when $\beta^{(N)}/N$ decreases. An extreme example is that when all tasks also share the same dynamics, leading to $\beta^{(N)} = \beta^{(1)}$. In this case, multi-task learning reduces to the batch setting and as the batch size increases, the iteration number decreases linearly in $N$.

## 4.3 IMPORTANT EXAMPLES OF MULTI-TASK PSRS

As shown in Section 3.2 and Theorem 1, for multi-task models with low $\eta$-bracketing number, i.e., satisfying $\beta^{(N)} < N\beta^{(1)}$, UMT-PSR exhibits better sample complexity than single-task learning. In this section, we provide example multi-task POMDPs and PSRs and show that their $\eta$-bracketing number satisfies the condition. Detailed proofs for these examples can be found in Appendix E.2.

**Muli-task POMDPs.** We consider tabular POMDPs, which is a classic subclass of PSRs. Specifically, the dynamics in POMDPs consist of $H$ transition distributions $\{\mathbb{T}_h : \mathcal{S} \times \mathcal{A} \times \mathcal{S} \rightarrow [0, 1]\}_{h=1}^{H}$, and $H$ emission distributions $\{\mathbb{O}_h : \mathcal{S} \times \mathcal{O} \rightarrow [0, 1]\}_{h=1}^{H}$, where $\mathcal{S}$ is a finite state space. The states capture the entire system information, but are not directly observable. In POMDPs, at each step $h$, if the current system state is $s_h$, the agent observes $o_h$ with probability $\mathbb{O}_h(o_h|s_h)$. Then, if the agent takes an action $a_h$ based on previous observations $o_h, \ldots, o_1$ and actions $a_{h-1}, \ldots, a_1$, the system state transits to $s_{h+1}$ with probability $\mathbb{T}_h(s_{h+1}|s_h, a_h)$. We use the notation $\mathrm{P_{po}} = (\mathcal{O}, \mathcal{A}, H, \mathcal{S}, \mathbb{T}, \mathbb{O}, R)$ to represent a POMDP instance. Note that the tuple $(\mathcal{S}, \mathbb{T}, \mathbb{O})$ in POMDPs determine the general dynamics $\mathbb{P}$ in PSRs. If all tasks share the same state, observation, and action spaces, then $\mathrm{P_{po}^n} = (\mathcal{O}, \mathcal{A}, H, \mathcal{S}, \mathbb{T}^n, \mathbb{O}^n, R)$ represents the model of task $n$.

**Example 1** (Multi-task POMDP with common transition kernels). *All tasks (i.e., all POMDPs) share the same transition kernel, i.e., there exists a set of transition distributions $\{\mathbb{T}_h^*\}_{h=1}^H$ such that $\mathbb{T}_h^n = \mathbb{T}_h^*$ for all $n \in [N]$ and $h \in [H]$. The emission distributions can be different. Such a scenario arises if the agent observes the same environment from different angles and hence receives different observations. Then, $\beta^{(N)}$ is at most $O(H(|\mathcal{S}|^2|\mathcal{A}| + |\mathcal{S}||\mathcal{O}|N) \log \frac{H|\mathcal{O}||\mathcal{A}||\mathcal{S}|}{\eta})$, whereas the single task $\beta^{(1)}$ is given by $O(H(|\mathcal{S}|^2|\mathcal{A}| + |\mathcal{S}||\mathcal{O}|) \log \frac{H|\mathcal{O}||\mathcal{A}||\mathcal{S}|}{\eta})$. Clearly, $\beta^{(N)} < N\beta^{(1)}$.*

**Multi-task PSRs:** We next provide two example multi-task PSRs, in which tasks do not share common model parameters. In these examples, the similarities among tasks could alternatively be established via implicit connections and correlations in latent spaces, which reduce the complexity of the joint model class, hence the estimation margin and the sample complexity of algorithms significantly compared with separately learning each single task.

**Example 2** (Multi-task PSR with perturbed models). *Suppose there exist a latent base task $\mathrm{P_b}$, and a finite noisy perturbation space $\boldsymbol{\Delta}$. Each task $n \in [N]$ is a noisy perturbation of the latent base task and can be parameterized into two parts: the base task plus a task-specified noise term. Specifically, for each step $h \in [H]$ and task $n \in [N]$, any $(o, a) \in \mathcal{O} \times \mathcal{A}$, we have*

$$\mathbf{M}_h^n(o_h, a_h) = \mathbf{M}_h^{\mathrm{b}}(o_h, a_h) + \Delta_h^n(o_h, a_h), \quad \Delta_h^n \in \boldsymbol{\Delta}.$$

*Such a multi-task PSR satisfies that $\beta^{(N)} = \tilde{O}(r^2|\mathcal{O}||\mathcal{A}|H^2 + HN \log |\boldsymbol{\Delta}|)$, whereas $\beta^{(1)}$ for a single task is given by $\tilde{O}(r^2|\mathcal{O}||\mathcal{A}|H^2)$. Clearly, $\beta^{(N)} \ll N\beta^{(1)}$ holds if $\log |\boldsymbol{\Delta}| \ll \tilde{O}(r^2|\mathcal{O}||\mathcal{A}|H)$, which can be easily satisfied for small-size perturbation environments. Hence, the multi-task PSR benefits from a significantly reduced sample complexity compared to single-task learning.*

**Example 3** (Multi-task PSR: Linear combination of core tasks). *Suppose that the multi-task PSR lies in the linear span of $m$ core tasks $\{\mathrm{P}_1, \ldots, \mathrm{P}_m\}$. Specifically, for each task $n \in [N]$, there exists a coefficient vector $\boldsymbol{\alpha}^n = (\alpha_1^n, \cdots, \alpha_m^n)^\top \in \mathbb{R}^m$ s.t. for any $h \in [H]$ and $(o_h, a_h) \in \mathcal{O} \times \mathcal{A}$,*

$$\phi_h^n(o_h, a_h) = \sum_{l=1}^m \alpha_l^n \phi_h^l(o_h, a_h), \quad \mathbf{M}_h^n(o_h, a_h) = \sum_{l=1}^m \alpha_l^n \mathbf{M}_h^l(o_h, a_h).$$

*For regularization, we assume $0 \leq \alpha_l^n$ for all $l \in [m]$ and $n \in [N]$, and $\sum_{l=1}^m \alpha_l^n = 1$ for all $n \in [N]$. It can be shown that $\beta^{(N)} = O(m(r^2|\mathcal{O}||\mathcal{A}|H^2 + N))$, whereas $\beta^{(1)} = \tilde{O}(r^2|\mathcal{O}||\mathcal{A}|H^2)$. Clearly, $\beta^{(N)} \ll N\beta^{(1)}$ holds if $m \leq \min\{N, r^2|\mathcal{O}||\mathcal{A}|H^2\}$, which is satisfied in practice.*

## 5 DOWNSTREAM LEARNING FOR PSRs

In downstream learning, the agent is assigned a new task $\mathrm{P}_0 = (\mathcal{O}, \mathcal{A}, H, \mathbb{P}_{\theta_0^*}, R_0)$, where $\theta_0^* \in \Theta_0^{\mathrm{u}}$, and $\Theta_0^{\mathrm{u}}$ is defined in Section 3.4. As explained in Section 3.4, upstream and downstream tasks are connected via the **similarity constraint** $\mathtt{C}(\theta_0, \theta_1^*, \ldots \theta_N^*) \leq \mathbf{0}$. Therefore, the agent can use the estimated model parameter $\bar{\theta}_1, \ldots, \bar{\theta}_N$ in the upstream to construct an empirical candidate model class for the downstream task as $\hat{\Theta}_0^{\mathrm{u}} = \{\theta_0 \in \Theta | \mathtt{C}(\theta_0, \bar{\theta}_1, \ldots, \bar{\theta}_N) \leq 0\}$. Then for downstream learning, we adopt the standard OMLE (Liu et al., 2022; Chen et al., 2022) for the model class $\hat{\Theta}_0^{\mathrm{u}}$.

The sample complexity of downstream learning will be determined by the bracketing number of $\hat{\Theta}_0^{\mathrm{u}}$, which is nearly the same as that of the ground truth $\Theta_0^{\mathrm{u}}$. Since the similarity constraint will significantly reduces the complexity of the model parameter space, the bracketing number of $\hat{\Theta}_0^{\mathrm{u}}$ should be much smaller than that of the original parameter space $\Theta$. In this way, the downstream can benefit from the upstream learning with reduced sample complexity. In the following subsections, we first characterize the performance guarantee for downstream learning in terms of the bracketing number of $\hat{\Theta}_0^{\mathrm{u}}$, and then show that the similarity constraint reduces the bracketing number for the examples given in Section 4.3.

### 5.1 THEORETICAL GUARANTEE FOR DOWNSTREAM LEARNING

One main challenge in the downstream learning is that the true model may not lie in $\hat{\Theta}_0^{\mathrm{u}}$. To handle this, we employ Rényi divergence to measure the "distance" from the model class to the true model as follows, mainly because its unique advantage under the MLE oracle: the Rényi divergence of order $\alpha$ with $\alpha \geq 1$ serves as an upper bound on the TV distance and the KL divergence, and thus has more robust performance.

**Definition 3.** *Fix $\alpha > 1$. The approximation error of $\hat{\Theta}_0^{\mathrm{u}}$ under $\alpha$-Rényi divergence is defined as*
$$\mathrm{e}_\alpha(\hat{\Theta}_0^{\mathrm{u}}) = \min_{\theta_0 \in \hat{\Theta}_0^{\mathrm{u}}} \max_{\pi \in \Pi} \mathrm{D}_{\mathrm{R},\alpha}(\mathbb{P}_{\theta_0^*}^\pi, \mathbb{P}_{\theta_0}^\pi).$$

**Theorem 2.** *Fix $\alpha > 1$. Let $\epsilon_0 = \mathrm{e}_\alpha(\hat{\Theta}_0^{\mathrm{u}})$, $\beta_0 = c_0(\log \mathcal{N}_\eta(\hat{\Theta}_0^{\mathrm{u}}) + \epsilon_0 KH + (\frac{\mathbf{1}_{\{\epsilon_0 \neq 0\}}}{\alpha-1} + 1)\log\frac{KH}{\delta})$ for some large $c_0$, where $\eta \leq \frac{1}{KH}$. Under Assumption 1, with probability at least $1 - \delta$, the output of Algorithm 2 satisfies that*

$$\max_{\pi \in \Pi} \mathrm{D}_{\mathrm{TV}}\left(\mathbb{P}_{\bar{\theta}_0}^\pi, \mathbb{P}_{\theta_0^*}^\pi\right) \leq \tilde{O}\left(\frac{Q_A}{\gamma}\sqrt{\frac{r|\mathcal{A}|H\beta_0}{K}} + \sqrt{\epsilon_0}\right).$$

**Benefits of downstream transfer learning:** Theorem 2 shows that when $\epsilon_0 < \epsilon^2/4$, with sample complexity at most $\tilde{O}(\frac{rQ_A^2|\mathcal{A}|H\beta_0}{\gamma^2\epsilon^2})$, OMLE identifies an $\epsilon$-optimal policy for the downstream task. As a comparison, the best known sample complexity for single-task PSR RL without transfer learning is $\tilde{O}(\frac{rQ_A^2|\mathcal{A}|H\beta}{\gamma^2\epsilon^2})$, where $\beta = \tilde{O}(\log\mathcal{N}_\eta(\Theta))$ (Chen et al., 2022). It is clear that as long as $\beta_0 < \beta$, then downstream learning enjoys transfer benefit in the sample complexity.

Notably, in the realizable case when $\epsilon_0 = 0$, i.e. $\theta_0^* \in \hat{\Theta}_0^{\mathrm{u}}$, we must have $\beta_0 = \tilde{O}(\log\mathcal{N}_\eta(\hat{\Theta}_0^{\mathrm{u}})) \leq \beta$, since $\hat{\Theta}_0^{\mathrm{u}} \subset \Theta$. In the non-realizable case when $\epsilon_0 > 0$, compared to the realizable case, the estimation error of $\bar{\theta}_0$ has an additive factor of $\tilde{O}(\sqrt{\epsilon_0} + \sqrt{1/(K(\alpha-1))})$ after hiding system parameters. We remark that this factor shrinks if the approximation error of $\hat{\Theta}_0^{\mathrm{u}}$ decreases and the order of Rényi divergence grows, which coincide with the intuition.

## 5.2 EXAMPLES IN DOWNSTREAM LEARNING TASKS

We revisit the examples presented in upstream multi-task learning, specifically Examples 1 to 3, and subsequently extend their application in downstream tasks under the realizable setting. With the prior knowledge obtained from upstream learning, these examples exhibit reduced $\eta$-bracketing number, and hence benefit in the sample efficiency. Detailed proofs are in Appx. E.3.

**Example 1** (Multi-task POMDP with Common transition kernels). *Suppose $\hat{\mathbb{T}}$ is the output from UMT-PSR. In this case, the downstream $\hat{\Theta}_0^{\mathrm{u}}$ is constructed by combining $\hat{\mathbb{T}}$ and all possible emission distributions. Then $\beta_0 = \tilde{O}(H|\mathcal{S}||\mathcal{O}|)$. However, for POMDP without prior knowledge, $\beta = \tilde{O}(H(|\mathcal{S}|^2|\mathcal{A}| + |\mathcal{S}||\mathcal{O}|))$. Clearly, $\beta_0 \leq \beta$, indicating the benefit of downstream learning.*

For PSRs without prior knowledge, we have $\beta^{\mathrm{PSR}} = \tilde{O}(r^2|\mathcal{O}||\mathcal{A}|H^2)$.

**Example 2** (Multi-task PSR with perturbed models). *The downstream task $\mathrm{P}_0$ is a noisy perturbation of a base task $\mathrm{P}_b$. Specifically, for each step $h \in [H]$, any $(o,a) \in \mathcal{O} \times \mathcal{A}$, we have*
$$\phi_H^0 = \phi_H^{\mathrm{b}}, \mathbf{M}_h^0(o_h, a_h) = \mathbf{M}_h^{\mathrm{b}}(o_h, a_h) + \Delta_h^0(o_h, a_h), \quad \Delta_h^0 \in \boldsymbol{\Delta}.$$

*Then, $\beta_0 = \tilde{O}(H\log|\boldsymbol{\Delta}|)$, which is much lower than $\beta^{\mathrm{PSR}}$ if $\log|\boldsymbol{\Delta}| \ll \tilde{O}(r^2|\mathcal{O}||\mathcal{A}|H)$.*

**Example 3** (Multi-task PSR: Linear combination of core tasks). *The downstream task $\mathrm{P}_0$ lies in the linear span of $L$ upstream tasks (e.g. the firs $L$ source tasks). Specifically, there exists a coefficient vector $\boldsymbol{\alpha}^0 = (\alpha_1^0, \cdots, \alpha_L^0)^\top \in \mathbb{R}^L$ s.t. for any $h \in [H]$ and $(o_h, a_h) \in \mathcal{O} \times \mathcal{A}$,*
$$\phi_H^0 = \sum_{l=1}^L \alpha_l^0 \phi_H^l, \quad \mathbf{M}_h^0(o_h, a_h) = \sum_{l=1}^L \alpha_l^0 \mathbf{M}_h^l(o_h, a_h).$$

*For regularization, we assume $0 \leq \alpha_l^0$ for all $l \in [L]$, and $\sum_{l=1}^L \alpha_l^0 = 1$. Then $\beta_0 = \tilde{O}(LH)$, which is much smaller than $\beta^{\mathrm{PSR}}$ if $L \leq \min\{N, r^2|\mathcal{O}||\mathcal{A}|H^2\}$.*

## 6 CONCLUSION

In this paper, we study multi-task learning on general non-markovian low-rank decision making problems. Given that all tasks share the same observation and action spaces, using the approach of PSRs, we theoretically characterize that multi-task learning presents benefit over single-task learning if the joint model class of PSRs has a smaller $\eta$-bracketing number. We also provide specific example multi-task PSRs with small $\eta$-bracketing numbers. Then, with prior knowledge from the upstream, we show that downstream learning is more efficient than learning from scratch.

ACKNOWLEDGMENTS

The work of R. Huang and J. Yang was supported in part by the U.S. National Science Foundation under the grants CNS-1956276, CNS-2003131 and CNS-2030026. The work of Y. Liang was supported in part by the U.S. National Science Foundation under the grants ECCS-2113860, DMS-2134145, and CNS-2112471. The work of Y. Cheng and V. Tan was supported by the Singapore Ministry of Education Academic Research Fund (AcRF) Tier 2 under grant number A-8000423-00-00 and the Singapore Ministry of Education (AcRF) Tier 1 under grant number A-8000189-01-00.

IMPACT STATEMENT

This paper presents work whose goal is to advance the field of Machine Learning. There are many potential societal consequences of our work, none which we feel must be specifically highlighted here.

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

# Supplementary Materials

## A  MULTI-TASK MLE ANALYSIS

The following lemma shows that the true model lies in the confidence set with high probability.

**Proposition 1** (Confidence Set). *For all $k \in [K]$, and for any $\boldsymbol{\theta} = (\theta_1, \ldots, \theta_N) \in \boldsymbol{\Theta}$, let $\eta \leq 1/(NKH)$, $\beta^{(N)} = c \log\left(\mathcal{N}_\eta(\boldsymbol{\Theta})NKH/\delta\right)$ for some $c \geq 0$. With probability at least $1 - \delta$, for any $k \in [K]$, we have*

$$\sum_{n \leq N} \sum_{t \leq k} \sum_{h \leq H} \log \mathbb{P}^{\nu_h^{\pi^{n,t}}}_{\theta_n^*}(\tau_H^{n,t,h}) \geq \sum_{n \leq N} \sum_{t \leq k} \sum_{h \leq H} \log \mathbb{P}^{\nu_h^{\pi^{n,t}}}_{\theta_n}(\tau_H^{n,t,h}) - \beta^{(N)}. \tag{4}$$

*Proof.* Consider a set of $\eta$-brackets, denoted by $\boldsymbol{\Theta}_\eta$, that covers $\boldsymbol{\Theta}$. For any $\boldsymbol{\theta}$, we can find two measures in $\boldsymbol{\Theta}_\eta$ parameterized by $\underline{\boldsymbol{\theta}}$ and $\overline{\boldsymbol{\theta}}$ such that

$$\forall n, \tau_H, \pi, \quad \mathbb{P}^\pi_{\overline{\theta}_n}(\tau_H) \geq \mathbb{P}^\pi_{\theta_n}(\tau_H) \geq \mathbb{P}^\pi_{\underline{\theta}_n}(\tau_H), \tag{5}$$

$$\forall n, \quad \mathrm{D_{TV}}\left(\mathbb{P}^\pi_{\overline{\theta}_n}(\tau_H), \mathbb{P}^\pi_{\underline{\theta}_n}(\tau_H)\right) \leq \eta.$$

Note that the above two inequalities imply that $\sum_{\tau_H} \mathbb{P}^\pi_{\overline{\theta}_n}(\tau_H) \leq \eta + \sum_{\tau_H} \mathbb{P}^\pi_{\underline{\theta}_n}(\tau_H) \leq 1 + \eta$.

Then, we have

$$\mathbb{E}\left[\exp\left(\sum_n \sum_{t=1}^k \sum_h \log \frac{\mathbb{P}^{\nu_h^{\pi^{n,t}}}_{\overline{\theta}_n}(\tau_H^{n,t,h})}{\mathbb{P}^{\nu_h^{\pi^{n,t}}}_{\theta_n^*}(\tau_H^{n,t,h})}\right)\right]$$

$$= \mathbb{E}\left[\exp\left(\sum_n \sum_{t=1}^{k-1} \sum_h \log \frac{\mathbb{P}^{\nu_h^{\pi^{n,t}}}_{\overline{\theta}_n}(\tau_H^{n,t,h})}{\mathbb{P}^{\nu_h^{\pi^{n,t}}}_{\theta_n^*}(\tau_H^{n,t,h})}\right) \mathbb{E}\left[\prod_n \prod_h \frac{\mathbb{P}^{\nu_h^{\pi^{n,k}}}_{\overline{\theta}_n}(\tau_H^{n,k,h})}{\mathbb{P}^{\nu_h^{\pi^{n,k}}}_{\theta_n^*}(\tau_H^{n,k,h})}\right]\right]$$

$$= \mathbb{E}\left[\exp\left(\sum_n \sum_{t=1}^{k-1} \sum_h \log \frac{\mathbb{P}^{\nu_h^{\pi^{n,t}}}_{\overline{\theta}_n}(\tau_H^{n,t,h})}{\mathbb{P}^{\nu_h^{\pi^{n,t}}}_{\theta_n^*}(\tau_H^{n,t,h})}\right) \prod_n \prod_h \sum_{\tau_H^{n,k,h}} \mathbb{P}^{\nu_h^{\pi^{n,k}}}_{\overline{\theta}_n}(\tau_H^{n,k,h})\right]$$

$$\leq \mathbb{E}\left[\exp\left(\sum_n \sum_{t=1}^{k-1} \sum_h \log \left[\frac{\mathbb{P}^{\nu_h^{\pi^{n,t}}}_{\overline{\theta}_n}(\tau_H^{n,t,h})}{\mathbb{P}^{\nu_h^{\pi^{n,t}}}_{\theta_n^*}(\tau_H^{n,t,h})}\right]\right) \prod_n \prod_h \left(1 + \frac{1}{NKH}\right)\right]$$

$$\leq \left(1 + \frac{1}{NKH}\right)^{NKH} \leq e.$$

Therefore, by Chernoff type bound, we have

$$\mathbb{P}\left(\sum_{\substack{t \leq k, h \leq H \\ n \leq N}} \log \frac{\mathbb{P}^{\nu_h^{\pi^{n,t}}}_{\overline{\theta}_n}(\tau_H^{n,t,h})}{\mathbb{P}^{\nu_h^{\pi^{n,t}}}_{\theta_n^*}(\tau_H^{n,t,h})} \geq \log(1/\delta)\right) \leq \frac{\mathbb{E}\left[\exp\left(\sum_{\substack{t \leq k, h \leq H \\ n \leq N}} \log \frac{\mathbb{P}^{\nu_h^{\pi^{n,t}}}_{\overline{\theta}_n}(\tau_H^{n,t,h})}{\mathbb{P}^{\nu_h^{\pi^{n,t}}}_{\theta_n^*}(\tau_H^{n,t,h})}\right)\right]}{1/\delta} \leq e\delta.$$

Taking a union bound over all $(\overline{\theta}, k) \in \overline{\boldsymbol{\Theta}}_\epsilon \times [K]$ and rescaling $\delta$, we have for any $\boldsymbol{\theta} \in \boldsymbol{\Theta}$,

$$\mathbb{P}\left(\forall \overline{\boldsymbol{\theta}} \in \boldsymbol{\Theta}_\eta, k \in [K], \sum_{\substack{t \leq k, h \leq H \\ n \leq N}} \log \frac{\mathbb{P}^{\nu_h^{\pi^{n,t}}}_{\overline{\theta}_n}(\tau_H^{n,t,h})}{\mathbb{P}^{\nu_h^{\pi^{n,t}}}_{\theta_n^*}(\tau_H^{n,t,h})} \geq \log(eK\mathcal{N}_\eta(\boldsymbol{\Theta})/\delta)\right) \leq \delta.$$

The proof is finished by noting that $\overline{\boldsymbol{\theta}}$ is an optimistic measure (see Equation (5)). $\qquad\square$

The following lemma establishes the relationship between the Hellinger-squared distance and the difference of log likelihood functions between true parameters and any possible parameters from the model class.

**Proposition 2.** *Let $\eta \leq 1/(N^2 K^2 H^2)$, $\beta^{(N)} = c \log\left(\mathcal{N}_\eta(\boldsymbol{\Theta})NKH/\delta\right)$ for some $c \geq 0$. Then, with probability at least $1 - \delta$, we have, for any $\boldsymbol{\theta} = (\theta_1, \ldots, \theta_N) \in \boldsymbol{\Theta}$, the following inequality holds.*

$$
\sum_{\substack{t \leq k, h \leq H \\ n \leq N}} \mathtt{D}_{\mathtt{H}}^2\left(\mathbb{P}_{\theta_n}^{\nu_h^{\pi^{n,t}}}, \mathbb{P}_{\theta_n^*}^{\nu_h^{\pi^{n,t}}}\right) \leq \sum_{\substack{t \leq k, h \leq H \\ n \leq N}} \log \frac{\mathbb{P}_{\theta_n^*}^{\nu_h^{\pi^{n,t}}}(\tau_H^{n,t,h})}{\mathbb{P}_{\theta_n}^{\nu_h^{\pi^{n,t}}}(\tau_H^{n,t,h})} + \beta^{(N)}.
$$

*Proof.* By the definition of $\eta$-bracket, for any multi-task parameter $\boldsymbol{\theta}$, we can find $\overline{\boldsymbol{\theta}}$ within a finite set of $\eta$-brackets such that $\sum_{\tau_H}\left|\mathbb{P}_{\theta_n}^\pi(\tau_H) - \mathbb{P}_{\overline{\theta}_n}^\pi(\tau_H)\right| \leq \eta$. Then, for any $n$ and $\pi$, we have

$$
\begin{aligned}
&\mathtt{D}_{\mathtt{H}}^2(\mathbb{P}_{\theta_n}^\pi, \mathbb{P}_{\theta_n^*}^\pi) \\
&= 1 - \sum_{\tau_H} \sqrt{\mathbb{P}_{\theta_n}^\pi(\tau_H)\mathbb{P}_{\theta_n^*}^\pi(\tau_H)} \\
&= 1 - \sum_{\tau_H} \sqrt{\mathbb{P}_{\overline{\theta}_n}^\pi(\tau_H)\mathbb{P}_{\theta^*}^\pi(\tau_H) + \left(\mathbb{P}_{\theta_n}^\pi(\tau_H) - \mathbb{P}_{\overline{\theta}_n}^\pi(\tau_H)\right)\mathbb{P}_{\theta_n^*}^\pi(\tau_H)} \\
&\overset{(i)}{\leq} 1 - \sum_{\tau_H} \sqrt{\mathbb{P}_{\overline{\theta}_n}^\pi(\tau_H)\mathbb{P}_{\theta_n^*}^\pi(\tau_H)} + \sum_{\tau_H} \sqrt{\left|\mathbb{P}_{\theta_n}^\pi(\tau_H) - \mathbb{P}_{\overline{\theta}_n}^\pi(\tau_H)\right|\mathbb{P}_{\theta_n^*}^\pi(\tau_H)} \\
&\overset{(ii)}{\leq} -\log \mathbb{E}_{\tau_H \sim \mathbb{P}_{\theta_n^*}^\pi(\cdot)} \sqrt{\frac{\mathbb{P}_{\overline{\theta}_n}^\pi(\tau_H)}{\mathbb{P}_{\theta_n^*}^\pi(\tau_H)}} + \sqrt{\sum_{\tau_H} \left|\mathbb{P}_{\overline{\theta}_n}^\pi(\tau_H) - \mathbb{P}_{\theta_n}^\pi(\tau_H)\right|} \\
&\overset{(iii)}{\leq} -\log \mathbb{E}_{\tau_H \sim \mathbb{P}_{\theta_n^*}^\pi(\cdot)} \sqrt{\frac{\mathbb{P}_{\overline{\theta}_n}^\pi(\tau_H)}{\mathbb{P}_{\theta_n^*}^\pi(\tau_H)}} + \sqrt{\eta},
\end{aligned}
\tag{6}
$$

where $(i)$ follows from the fact that $\sqrt{a+b} \leq \sqrt{|a|} + \sqrt{|b|}$, $(ii)$ follows from the fact that $1 - x \leq \log x$ for $x \geq 0$ (the first term) and the Cauchy-Schwarz inequality (the second term), and $(iii)$ follows from the definition of $\eta$-bracket.

Then, for any fixed $\overline{\theta}_n$, we have

$$
\begin{aligned}
&\mathbb{E}\left[\exp\left(\frac{1}{2}\sum_{n \leq N}\sum_{t \leq k}\sum_{h \leq H}\log\frac{\mathbb{P}_{\overline{\theta}_n}^{\nu_h^{\pi^{n,t}}}(\tau_H^{n,t,h})}{\mathbb{P}_{\theta_h^*}^{\nu_h^{\pi^{n,t}}}(\tau_H^{n,t,h})} - \sum_n\sum_{\pi \in \Pi_n^k}\log\mathbb{E}_{\tau_H \sim \mathbb{P}_{\theta_n^*}^\pi(\cdot)}\sqrt{\frac{\mathbb{P}_{\overline{\theta}_n}^\pi(\tau_H)}{\mathbb{P}_{\theta_n^*}^\pi(\tau_H)}}\right)\right] \\
&= \frac{\mathbb{E}\left[\prod_{n \leq N}\prod_{t \leq k}\prod_{h \leq H}\sqrt{\frac{\mathbb{P}_{\overline{\theta}_n}^{\nu_h^{\pi^{n,t}}}(\tau_H)}{\mathbb{P}_{\theta_n^*}^{\nu_h^{\pi^{n,t}}}(\tau_H)}}\right]}{\mathbb{E}\left[\prod_{n \leq N}\prod_{t \leq k}\prod_{h \leq H}\sqrt{\frac{\mathbb{P}_{\overline{\theta}_n}^{\nu_h^{\pi^{n,t}}}(\tau_H)}{\mathbb{P}_{\theta_n^*}^{\nu_h^{\pi^{n,t}}}(\tau_H)}}\right]} \\
&= 1.
\end{aligned}
\tag{7}
$$

Hence, by taking union bound over the finite set of $\eta$-brackets and $k \in [K]$, with probability at least $1 - \delta$, we have for any $k \in [K]$

$$
\sum_{n \leq N}\sum_{t \leq k}\sum_{h \leq H}\mathtt{D}_{\mathtt{H}}^2(\mathbb{P}_{\theta_n}^{\nu_h^{\pi^{n,t}}}, \mathbb{P}_{\theta_n^*}^{\nu_h^{\pi^{n,t}}})
$$

$$\overset{(i)}{\leq} \sum_{n \leq N} \sum_{t \leq k} \sum_{h \leq H} - \log \underset{\tau_H \sim \mathbb{P}_{\theta_n^*}^{\nu_h^{\pi^{n,t}}}(\cdot)}{\mathbb{E}} \sqrt{\frac{\mathbb{P}_{\bar{\theta}_n}^{\nu_h^{\pi^{n,t}}}(\tau_H)}{\mathbb{P}_{\theta_n^*}^{\nu_h^{\pi^{n,t}}}(\tau_H)}} + NKH\sqrt{\eta}$$

$$\overset{(ii)}{\leq} NKH\sqrt{\eta} + \frac{1}{2} \sum_{n \leq N} \sum_{t \leq k} \sum_{h \leq H} \log \frac{\mathbb{P}_{\theta_n^*}^{\nu_h^{\pi^{n,t}}}(\tau_H^{n,t,h})}{\mathbb{P}_{\bar{\theta}_n}^{\nu_h^{\pi^{n,t}}}(\tau_H^{n,t,h})} + \log \frac{K\mathcal{N}_\eta(\Theta)}{\delta}$$

$$\overset{(iii)}{\leq} 1 + \frac{1}{2} \sum_{n \leq N} \sum_{t \leq k} \sum_{h \leq H} \log \frac{\mathbb{P}_{\theta_n^*}^{\nu_h^{\pi^{n,t}}}(\tau_H^{n,t,h})}{\mathbb{P}_{\bar{\theta}_n}^{\nu_h^{\pi^{n,t}}}(\tau_H^{n,t,h})} + \log \frac{K\mathcal{N}_\eta(\Theta)}{\delta}.$$

where $(i)$ follows from Equation (6), $(ii)$ follows from Equation (7), the Chernoff's method and the union bound, and $(iii)$ follows from that $\eta \leq 1/(N^2K^2H^2)$.

The proof is finished by noting that $\bar{\theta}_n$ is an optimistic measure.

$\square$

# B PROPERTIES OF PSRs

First, for any model $\theta = \{\phi_h, \mathbf{M}_h(o_h, a_h)\}$, we have the following identity

$$\mathbf{M}_h(o_h, a_h)\bar{\psi}(\tau_{h-1}) = \mathbb{P}_\theta(o_h|\tau_{h-1})\bar{\psi}(\tau_h). \tag{8}$$

The following proposition is adapted from Lemma C.3 in Liu et al. (2022) and Proposition 1 in Huang et al. (2023).

**Proposition 3** (TV-distance $\leq$ Estimation error). *For any task $n \in [N]$, policy $\pi$, and any two parameters $\theta, \theta' \in \Theta$, we have*

$$\mathtt{D}_{\mathtt{TV}}\left(\mathbb{P}_{\theta'}^\pi, \mathbb{P}_\theta^\pi\right) \leq \sum_{h=1}^H \sum_{\tau_H} \left|\mathbf{m}'(\omega_h)^\top \left(\mathbf{M}_h'(o_h, a_h) - \mathbf{M}_h(o_h, a_h)\right)\psi(\tau_{h-1})\right| \pi(\tau_H).$$

# C PROOFS FOR UPSTREAM LEARNING: PROOF OF THEOREM 1

In this section, we first prove two lemmas, and then provide the proof for Theorem 1.

First, by the algorithm design and the construction of the confidence set, we have the following estimation guarantee.

**Lemma 1** (Estimation Guarantee in Upstream Learning). *Let $\eta \leq 1/(N^2K^2H^2)$, $\beta^{(N)} = c\log(\mathcal{N}_\eta(\Theta)NKH/\delta)$ for some $c \geq 0$. With probability at least $1 - \delta$, for any $k$ and any $\hat{\theta} = (\hat{\theta}_1 \ldots, \hat{\theta}_n) \in \mathcal{B}_k$, we have*

$$\sum_{n \leq N} \sum_{t \leq k-1} \sum_{h \leq H} \mathtt{D}_{\mathtt{H}}^2\left(\mathbb{P}_{\hat{\theta}_n}^{\nu_h^{\pi^{n,t}}}, \mathbb{P}_{\theta_n^*}^{\nu_h^{\pi^{n,t}}}\right) \leq 2\beta^{(N)}.$$

*Proof.* The proof follows directly by combining Proposition 1 and Proposition 2, and the optimality of the confidence set $\mathcal{B}_k$. $\square$

Then, we show that estimation error can be upper bounded by the norm of prediction features.

By Lemma G.3 in Liu et al. (2022), for any task $n$, and step $h$, we can find a projection $\mathbf{A}_h^n \in \mathbb{R}^{d_{h-1} \times r}$ such that

$$(i): \psi^{n,*}(\tau_{h-1}) = \mathbf{A}_h^n(\mathbf{A}_h^n)^\dagger \psi^{n,*}(\tau_{h-1}), \quad (ii): \|\mathbf{A}_h^n\|_1 \leq 1. \tag{9}$$

**Lemma 2.** *Let $\mathbf{A}_h^n \in \mathbb{R}^{d_{h-1} \times r}$ be the projector satisfying Equation* (9). *Fix $k \in [K]$. For any $\hat{\boldsymbol{\theta}}^k = (\hat{\theta}_1^k \ldots, \hat{\theta}_n^k) \in \mathcal{B}_k$ and any multi-task policy $\boldsymbol{\pi} = (\pi_1, \ldots, \pi_n)$, we have*

$$\sum_{n \leq N} \mathsf{D}_{\mathsf{TV}} \left( \mathbb{P}_{\hat{\theta}_n^k}^{\pi_n}, \mathbb{P}_{\theta_n^*}^{\pi_n} \right) \leq \sqrt{\sum_{n \leq N} \sum_{h \leq H} \mathbb{E}_{\tau_{h-1} \sim \mathbb{P}_{\theta_n^*}^{\pi_n}} \left[ \left\| (\mathbf{A}_h^n)^\dagger \bar{\psi}^{n,*}(\tau_{h-1}) \right\|_{(U_{k,h}^n)^{-1}}^2 \right]},$$

*where*

$$U_{k,h}^n = \lambda I + (\mathbf{A}_h^n)^\dagger \sum_{t<k} \mathbb{E}_{\tau_{h-1} \sim \mathbb{P}_{\theta_n^*}^{\pi^{n,t}}} \left[ \bar{\psi}^{n,*}(\tau_{h-1}) \bar{\psi}^{n,*}(\tau_{h-1})^\top \right] ((\mathbf{A}_h^n)^\dagger)^\top.$$

*Proof.* By Proposition 3, we have

$$\mathsf{D}_{\mathsf{TV}} \left( \mathbb{P}_{\hat{\theta}_n^k}^{\pi_n}, \mathbb{P}_{\theta_n^*}^{\pi_n} \right) \leq \sum_{h=1}^{H} \sum_{\tau_H} \left| \hat{\mathbf{m}}^{n,k}(\omega_h)^\top \left( \hat{\mathbf{M}}_h^{n,k}(o_h, a_h) - \mathbf{M}_h^{n,*}(o_h, a_h) \right) \psi^{n,*}(\tau_{h-1}) \right| \pi_n(\tau_H).$$

For ease of presentation, we fix a task index $n$. Index $\tau_{h-1}$ by $i$, $\omega_{h-1}$ by $j$. Denote $(\mathbf{A}_h^n)^\dagger \bar{\psi}^{n,*}(\tau_{h-1})$ by $x_i^n$, $\hat{\mathbf{m}}^{n,k}(\omega_h)^\top \left( \hat{\mathbf{M}}_h^{n,k}(o_h, a_h) - \mathbf{M}_h^{n,*}(o_h, a_h) \right) \mathbf{A}_h^n \pi_n(\omega_{h-1}|\tau_{h-1})$ by $(w_{j|i}^n)^\top$.

Then, we have

$$\sum_{\tau_H} \left| \hat{\mathbf{m}}^{n,k}(\omega_h)^\top \left( \hat{\mathbf{M}}_h^{n,k}(o_h, a_h) - \mathbf{M}^{n,*}(o_h, a_h) \right) \psi^{n,*}(\tau_{h-1}) \right| \pi_n(\tau_H)$$

$$\overset{(i)}{=} \sum_{\tau_H} \left| \hat{\mathbf{m}}^{n,k}(\omega_h)^\top \left( \hat{\mathbf{M}}_h^{n,k}(o_h, a_h) - \mathbf{M}^{n,*}(o_h, a_h) \right) \bar{\psi}^{n,*}(\tau_{h-1}) \pi_n(\omega_{h-1}|\tau_{h-1}) \right| \mathbb{P}_{\theta_n^*}^{\pi_n}(\tau_{h-1})$$

$$= \sum_i \mathbb{P}_{\theta_n^*}^{\pi_n}(i) \sum_j \left| (w_{j|i}^n)^\top x_i^n \right|$$

$$= \mathbb{E}_{i \sim \mathbb{P}_{\theta_n^*}^{\pi_n}} \left[ (x_i^n)^\top \left( \sum_j w_{j|i}^n \mathtt{sgn}((w_{j|i}^n)^\top x_i^n) \right) \right]$$

$$\overset{(ii)}{\leq} \mathbb{E}_{i \sim \mathbb{P}_{\theta_n^*}^{\pi_n}} \left[ \|x_i^n\|_{(U_{k,h-1}^n)^{-1}} \left\| \sum_j w_{j|i}^n \mathtt{sgn}((w_{j|i}^n)^\top x_i^n) \right\|_{U_{k,h-1}^n} \right],$$

where $(i)$ follows from the property of the projection $\mathbf{A}_h^n$ and definition of the prediction feature $\bar{\psi}^{n,*}$, and $(ii)$ is due to the Cauchy's inequality.

Fix an index $i = i_0$. We aim to analyze $\left\| \sum_j w_{j|i_0}^n \mathtt{sgn}((w_{j|i_0}^n)^\top x_{i_0}^n) \right\|_{U_{k,h-1}^n}$. We have

$$\left\| \sum_j w_{j|i_0}^n \mathtt{sgn}((w_{j|i_0}^n)^\top x_{i_0}^n) \right\|_{U_{k,h-1}^n}^2$$

$$= \underbrace{\lambda \left\| \sum_j \mathtt{sgn}((w_{j|i_0}^n)^\top x_{i_0}^n) w_{j|i_0}^n \right\|_2^2}_{I_1} + \underbrace{\sum_{t<k} \mathbb{E}_{i \sim \mathbb{P}_{\theta_n^*}^{\nu_h^{\pi^{n,t}}}} \left( \sum_j \mathtt{sgn}((w_{j|i_0}^n)^\top x_{i_0}^n)(w_{j|i_0}^n)^\top x_i^n \right)^2}_{I_2}.$$

For the first term $I_1$, we have

$$\sqrt{I_1} = \sqrt{\lambda} \max_{x \in \mathbb{R}^r : \|x\|_2 = 1} \left| \sum_j \mathtt{sgn}((w_{j|i_0}^n)^\top x_{i_0}^n)(w_{j|i_0}^n)^\top x \right|$$

$$\leq \sqrt{\lambda} \max_{x \in \mathbb{R}^r : \|x\|_2 = 1} \sum_{\omega_{h-1}} \pi_n(\omega_{h-1}|i_0) \left| \hat{\mathbf{m}}^{n,k}(\omega_h)^\top \left( \hat{\mathbf{M}}_h^{n,k}(o_h, a_h) - \mathbf{M}_h^{n,*}(o_h, a_h) \right) \mathbf{A}_h^n x \right|$$

$$
\leq \sqrt{\lambda} \max_{x\in\mathbb{R}^r:\|x\|_2=1} \sum_{\omega_{h-1}} \pi_n(\omega_{h-1}|i_0) \left|\hat{\mathbf{m}}^{n,k}(\omega_{h-1})^\top \mathbf{A}_h^n x\right|
$$

$$
+ \sqrt{\lambda} \max_{x\in\mathbb{R}^r:\|x\|_2=1} \sum_{\omega_{h-1}} \pi_n(\omega_{h-1}|i_0) \left|\hat{\mathbf{m}}^{n,k}(\omega_h)^\top \mathbf{M}_h^{n,*}(o_h,a_h)\mathbf{A}_h^n x\right|
$$

$$
\overset{(i)}{\leq} \frac{\sqrt{\lambda}}{\gamma} \max_{x\in\mathbb{R}^r:\|x\|_2=1} \|\mathbf{A}_h^n x\|_1 + \frac{\sqrt{\lambda}}{\gamma} \max_{x\in\mathbb{R}^r:\|x\|_2=1} \sum_{o_h,a_h} \pi_n(a_h|o_h,i_0)\|\mathbf{M}^{n,*}(o_h,a_h)\mathbf{A}_h^n x\|_1
$$

$$
\overset{(ii)}{\leq} \frac{2\sqrt{\lambda}r Q_A}{\gamma^2},
$$

where $(i)$ follows from Assumption 1, and $(ii)$ follows from the property of $\mathbf{A}_h^n$ stated before Lemma 2.

For the second term $I_2$, we have

$$
I_2 \leq \sum_{t<k} \mathbb{E}_{\tau_{h-1}\sim\mathbb{P}_{\theta_n^*}^{\pi_n,t}} \left(\sum_{\omega_{h-1}} \pi_n(\omega_{h-1}|i_0)\left|\hat{\mathbf{m}}^{n,k}(\omega_h)^\top \left(\hat{\mathbf{M}}_h^{n,k}(o_h,a_h)-\mathbf{M}_h^{n,*}(o_h,a_h)\right)\bar{\psi}^{n,*}(\tau_{h-1})\right|\right)^2
$$

$$
\leq \sum_{t<k} \mathbb{E}_{\tau_{h-1}\sim\mathbb{P}_{\theta_n^*}^{\pi_n,t}} \left(\sum_{\omega_{h-1}} \pi_n(\omega_{h-1}|i_0)\left|\hat{\mathbf{m}}^{n,k}(\omega_h)^\top \hat{\mathbf{M}}^{n,k}(o_h,a_h)\left(\bar{\psi}^{n,*}(\tau_{h-1})-\bar{\hat{\psi}}^{n,k}(\tau_{h-1})\right)\right|\right.
$$

$$
\left.+ \sum_{\omega_{h-1}} \pi_n(\omega_{h-1}|i_0)\left|\hat{\mathbf{m}}^{n,k}(\omega_h)^\top \left(\hat{\mathbf{M}}^{n,k}(o_h,a_h)\bar{\hat{\psi}}^{n,k}(\tau_{h-1})-\mathbf{M}^{n,*}(o_h,a_h)\bar{\psi}^{n,*}(\tau_{h-1})\right)\right|\right)^2
$$

$$
\overset{(a)}{\leq} \frac{1}{\gamma^2}\sum_{t<k} \mathbb{E}_{\tau_{h-1}\sim\mathbb{P}_{\theta_n^*}^{\pi_n,t}} \left[\left(\left\|\bar{\psi}^{n,*}(\tau_{h-1})-\bar{\hat{\psi}}^{n,k}(\tau_{h-1})\right\|_1\right.\right.
$$

$$
\left.\left.+ \sum_{o_h,a_h} \pi_n(a_h|o_h,i_0)\left\|\mathbb{P}_{\hat{\theta}_n^k}(o_h|\tau_{h-1})\bar{\hat{\psi}}^{n,k}(\tau_h)-\mathbb{P}_{\theta_n^*}(o_h|\tau_{h-1})\bar{\psi}^{n,*}(\tau_h)\right\|_1\right)^2\right]
$$

$$
\leq \frac{2}{\gamma^2}\sum_{t<k} \underbrace{\mathbb{E}_{\tau_{h-1}\sim\mathbb{P}_{\theta_n^*}^{\pi_n,t}} \left[\left\|\bar{\psi}^{n,*}(\tau_{h-1})-\bar{\hat{\psi}}^{n,k}(\tau_{h-1})\right\|_1^2\right]}_{I_{21}}
$$

$$
+ \frac{2}{\gamma^2}\sum_{t<k} \underbrace{\mathbb{E}_{\tau_{h-1}\sim\mathbb{P}_{\theta_n^*}^{\pi_n,t}} \left[\left(\sum_{o_h,a_h} \pi_n(a_h|o_h,i_0)\left\|\mathbb{P}_{\hat{\theta}_n^k}(o_h|\tau_{h-1})\bar{\hat{\psi}}^{n,k}(\tau_h)-\mathbb{P}_{\theta_n^*}(o_h|\tau_{h-1})\bar{\psi}^{n,*}(\tau_h)\right\|_1\right)^2\right]}_{I_{22}},
$$

where $(a)$ is due to Equation (8) and Assumption 1.

Recall that the $\ell$-th coordinate of a prediction feature $\bar{\psi}(\tau_{h-1})$ is the conditional probability of core test $\mathbf{o}_{h-1}^\ell$. Hence, for the term $I_{21}$, we have

$$
I_{21} = \mathbb{E}_{\tau_{h-1}\sim\mathbb{P}_{\theta_n^*}^{\pi_n,t}} \left[\left(\sum_{\ell=1}^{d_{h-1}^n} \left|\mathbb{P}_{\hat{\theta}_n^k}(\mathbf{o}_{h-1}^{n,\ell}|\tau_{h-1},\mathbf{a}_{h-1}^{n,\ell})-\mathbb{P}_{\theta_n^*}(\mathbf{o}_{h-1}^{n,\ell}|\tau_{h-1},\mathbf{a}_{h-1}^{n,\ell})\right|\right)^2\right]
$$

$$
\leq Q_A^2 \mathbb{E}_{\tau_{h-1}\sim\mathbb{P}_{\theta_n^*}^{\pi_n,t}} \left[\left(\mathbb{E}_{\mathbf{a}\sim\mathsf{u}_{\mathcal{Q}_{h-1}^{n,A}}} \sum_{\mathbf{o}_{h-1}} \left|\mathbb{P}_{\hat{\theta}_n^k}(\mathbf{o}_{h-1}|\tau_{h-1},\mathbf{a})-\mathbb{P}_{\theta_n^*}(\mathbf{o}_{h-1}|\tau_{h-1},\mathbf{a})\right|\right)^2\right]
$$

$$
= Q_A^2 \mathbb{E}_{\tau_{h-1}\sim\mathbb{P}_{\theta_n^*}^{\pi_n,t}} \mathsf{D}_{\mathrm{TV}}^2\left(\mathbb{P}_{\hat{\theta}_n^k}^{\mathsf{u}_{\mathcal{Q}_{h-1}^{n,A}}}(\omega_{h-1}|\tau_{h-1}),\mathbb{P}_{\theta_n^*}^{\mathsf{u}_{\mathcal{Q}_{h-1}^{n,A}}}(\omega_{h-1}|\tau_{h-1})\right)
$$

$$\overset{(a)}{\leq} Q_A^2 |\mathcal{A}| \mathop{\mathbb{E}}_{\tau_{h-2}, o_{h-1} \sim \mathbb{P}_{\theta_n^*}^{\pi^{n,t}}} \mathop{\mathbb{E}}_{a_{h-1} \sim \mathbf{u}_{\mathcal{A}}} \mathtt{D}_{\mathtt{H}}^2 \left( \mathbb{P}_{\hat{\theta}_n^k}^{\mathbf{u}_{\mathcal{Q}_{h-1}^{n,A}}} (\omega_{h-1}|\tau_{h-1}), \mathbb{P}_{\theta_n^*}^{\mathbf{u}_{\mathcal{Q}_{h-1}^{n,A}}} (\omega_{h-1}|\tau_{h-1}) \right)$$

$$\overset{(b)}{\leq} Q_A^2 |\mathcal{A}| \mathtt{D}_{\mathtt{H}}^2 \left( \mathbb{P}_{\hat{\theta}_n^k}^{\nu_{h-1}^{\pi^{n,t}}} (\tau_H), \mathbb{P}_{\theta_n^*}^{\nu_{h-1}^{\pi^{n,t}}} (\tau_H) \right),$$

where $(a)$ and $(b)$ follow from Lemma 7.

In addition, we can bound $I_{22}$ as follows.

$$I_{22} \leq 2 \mathop{\mathbb{E}}_{\tau_{h-1} \sim \mathbb{P}_{\theta_n^*}^{\pi^{n,t}}} \left( \sum_{o_h, a_h} \pi_n(a_h|o_h, \tau_{h-1}) \left| \mathbb{P}_{\hat{\theta}_n^k}(o_h|\tau_{h-1}) - \mathbb{P}_{\theta_n^*}(o_h|\tau_{h-1}) \right| \|\hat{\bar{\psi}}^{n,k}(\tau_h)\|_1 \right)^2$$

$$+ 2 \mathop{\mathbb{E}}_{\tau_{h-1} \sim \mathbb{P}_{\theta_n^*}^{\pi^{n,t}}} \left( \sum_{o_h, a_h} \pi_n(a_h|o_h, \tau_{h-1}) \mathbb{P}_{\theta_n^*}(o_h|\tau_{h-1}) \left\| \hat{\bar{\psi}}^{n,k}(\tau_h) - \bar{\psi}^{n,*}(\tau_h) \right\|_1 \right)^2$$

$$\overset{(i)}{\leq} 2 Q_A^2 \mathop{\mathbb{E}}_{\tau_{h-1} \sim \mathbb{P}_{\theta_n^*}^{\pi^{n,t}}} \mathtt{D}_{\mathtt{TV}}^2 \left( \mathbb{P}_{\hat{\theta}_n^k}(o_h|\tau_{h-1}), \mathbb{P}_{\theta^*}(o_h|\tau_{h-1}) \right)$$

$$+ 2 Q_A^2 \mathop{\mathbb{E}}_{\tau_{h-1} \sim \mathbb{P}_{\theta}^{\pi^{n,t}}} \mathop{\mathbb{E}}_{o_h \sim \mathbb{P}_{\theta_n^*}(\cdot|\tau_{h-1})} \mathop{\mathbb{E}}_{a_h \sim \pi_n} \mathtt{D}_{\mathtt{TV}}^2 \left( \mathbb{P}_{\hat{\theta}_n^k}^{\mathbf{u}_{\mathcal{Q}_h^{n,A}}} (\omega_h|\tau_h), \mathbb{P}_{\theta_n^*}^{\mathbf{u}_{\mathcal{Q}_h^{n,A}}} (\omega_h|\tau_h) \right)$$

$$\overset{(ii)}{\leq} 2 Q_A^2 |\mathcal{A}| \mathop{\mathbb{E}}_{\tau_{h-2}, o_{h-1} \sim \mathbb{P}_{\theta_n^*}^{\pi^{n,t}}} \mathop{\mathbb{E}}_{a_{h-1} \sim \mathbf{u}_{\mathcal{A}}} \mathtt{D}_{\mathtt{H}}^2 \left( \mathbb{P}_{\hat{\theta}_n^k}(o_h|\tau_{h-1}), \mathbb{P}_{\theta^*}(o_h|\tau_{h-1}) \right)$$

$$+ 2 Q_A^2 |\mathcal{A}| \mathop{\mathbb{E}}_{\tau_{h-1} \sim \mathbb{P}_{\theta}^{\pi^{n,t}}} \mathop{\mathbb{E}}_{o_h \sim \mathbb{P}_{\theta_n^*}(\cdot|\tau_{h-1})} \mathop{\mathbb{E}}_{a_h \sim \mathbf{u}_{\mathcal{A}}} \mathtt{D}_{\mathtt{H}}^2 \left( \mathbb{P}_{\hat{\theta}_n^k}^{\mathbf{u}_{\mathcal{Q}_h^{n,A}}} (\omega_h|\tau_h), \mathbb{P}_{\theta_n^*}^{\mathbf{u}_{\mathcal{Q}_h^{n,A}}} (\omega_h|\tau_h) \right)$$

$$\overset{(iii)}{\leq} 2 Q_A^2 |\mathcal{A}| \left( \mathtt{D}_{\mathtt{H}}^2 \left( \mathbb{P}_{\hat{\theta}_n^k}^{\nu_{h-1}^{\pi^{n,t}}} (\tau_H), \mathbb{P}_{\theta_n^*}^{\nu_{h-1}^{\pi^{n,t}}} (\tau_H) \right) + \mathtt{D}_{\mathtt{H}}^2 \left( \mathbb{P}_{\hat{\theta}_n^k}^{\nu_h^{\pi^{n,t}}} (\tau_H), \mathbb{P}_{\theta_n^*}^{\nu_h^{\pi^{n,t}}} (\tau_H) \right) \right),$$

where $(i)$ follows from that the coordinate of $\bar{\psi}$ takes the value on the conditional probability over core test, $(ii)$ and $(iii)$ follow from Lemma 7.

Substituting the upper bounds of $I_{21}$ and $I_{22}$ into $I_2$, we obtain that

$$I_2 \leq \frac{6 Q_A^2 |\mathcal{A}|}{\gamma^2} \sum_{t<k} \left( \mathtt{D}_{\mathtt{H}}^2 \left( \mathbb{P}_{\hat{\theta}_n^k}^{\nu_{h-1}^{\pi^{n,t}}} (\tau_H), \mathbb{P}_{\theta_n^*}^{\nu_{h-1}^{\pi^{n,t}}} (\tau_H) \right) + \mathtt{D}_{\mathtt{H}}^2 \left( \mathbb{P}_{\hat{\theta}_n^k}^{\nu_h^{\pi^{n,t}}} (\tau_H), \mathbb{P}_{\theta_n^*}^{\nu_h^{\pi^{n,t}}} (\tau_H) \right) \right).$$

Denote $\mathtt{D}_{\mathtt{H}}^2 \left( \mathbb{P}_{\hat{\theta}_n^k}^{\nu_h^{\pi^{n,t}}} (\tau_H), \mathbb{P}_{\theta_n^*}^{\nu_h^{\pi^{n,t}}} (\tau_H) \right)$ by $\zeta_{t,h}^n$. Therefore,

$$\mathtt{D}_{\mathtt{TV}} \left( \mathbb{P}_{\hat{\theta}_n^k}^{\pi_n}, \mathbb{P}_{\theta^*}^{\pi_n} \right) \leq \sum_h \mathbb{E}_{\tau_{h-1} \sim \mathbb{P}_{\theta_n^*}^{\pi_n}} \left[ \sqrt{C_\lambda + \sum_{t<k} \zeta_{t,h-1}^n + \sum_{t<k} \zeta_{t,h}^n} \left\| (\mathbf{A}_h^n)^\dagger \bar{\psi}^{n,*}(\tau_{h-1}) \right\|_{(U_{k,h}^n)^{-1}} \right].$$

Summing over $n$, we have

$$\sum_n \mathtt{D}_{\mathtt{TV}} \left( \mathbb{P}_{\hat{\theta}_n^k}^{\pi_n}, \mathbb{P}_{\theta^*}^{\pi_n} \right)$$

$$\leq \sum_n \sum_h \sqrt{C_\lambda + \sum_{t<k} \zeta_{t,h-1}^n + \sum_{t<k} \zeta_{t,h}^n} \mathbb{E}_{\tau_{h-1} \sim \mathbb{P}_{\theta_n^*}^{\pi_n}} \left[ \left\| (\mathbf{A}_h^n)^\dagger \bar{\psi}^{n,*}(\tau_{h-1}) \right\|_{(U_{k,h}^n)^{-1}} \right]$$

$$\leq \sqrt{NHC_\lambda + \sum_n \sum_h \sum_{t<k} (\zeta_{t,h-1}^n + \zeta_{t,h}^n)} \sqrt{\sum_n \sum_h \left( \mathbb{E}_{\tau_{h-1} \sim \mathbb{P}_{\theta_n^*}^{\pi_n}} \left[ \left\| (\mathbf{A}_h^n)^\dagger \bar{\psi}^{n,*}(\tau_{h-1}) \right\|_{(U_{k,h}^n)^{-1}} \right] \right)^2}$$

$$\overset{(i)}{\leq} \frac{Q_A \sqrt{|\mathcal{A}| \beta^{(N)}}}{\gamma} \sqrt{\sum_n \sum_h \left( \mathbb{E}_{\tau_{h-1} \sim \mathbb{P}_{\theta_n^*}^{\pi_n}} \left[ \left\| (\mathbf{A}_h^n)^\dagger \bar{\psi}^{n,*}(\tau_{h-1}) \right\|_{(U_{k,h}^n)^{-1}} \right] \right)^2},$$

where $(i)$ is due to the estimation guarantee Lemma 1.

$\square$

**Theorem 3** (Restatement of Theorem 1). *For any fixed $\delta \geq 0$, let $\Theta$ be the multi-task parameter space, $\beta^{(N)} = c_1(\log \frac{KHN}{\delta} + \log \mathcal{N}_\eta(\Theta))$, where $c_1 \geq 0$ and $\eta \leq \frac{1}{KHN}$. Then, under Assumption 1, with probability at least $1 - \delta$, UMT-PSR finds a multi-task model $\bar{\boldsymbol{\theta}} = (\bar{\theta}_1, \ldots, \bar{\theta}_N)$ such that*

$$\sum_{n=1}^{N} \max_{\pi^n} \mathsf{D}_{\mathsf{TV}} \left( \mathbb{P}_{\bar{\theta}_n}^{\pi^n}, \mathbb{P}_{\theta_n^*}^{\pi^n} \right) \leq \tilde{O} \left( \frac{Q_A}{\gamma} \sqrt{\frac{rH|\mathcal{A}|N\beta^{(N)}}{K}} \right). \tag{10}$$

*In addition, if $K = \frac{c_2 r|\mathcal{A}|Q_A^2 H\beta^{(N)}}{N\gamma^2\epsilon^2}$ for some $c_2$, UMT-PSR produces a multi-task policy $\overline{\boldsymbol{\pi}} = (\bar{\pi}^1, \ldots, \bar{\pi}^N)$ such that the average sub-optimality gap is at most $\epsilon$, i.e.*

$$\frac{1}{N} \sum_{n=1}^{N} \left( \max_{\pi} V_{\theta_n^*, R_n}^{\pi} - V_{\theta^*, R_n}^{\bar{\pi}^n} \right) \leq \epsilon. \tag{11}$$

*Proof.* Note that $\bar{\boldsymbol{\theta}} \in \boldsymbol{\mathcal{B}}_k$ for all $k \in [K+1]$. Therefore, by Lemma 2, we have

$$K \sum_n \mathsf{D}_{\mathsf{TV}} \left( \mathbb{P}_{\bar{\theta}_n}^{\pi_n}, \mathbb{P}_{\theta_n^*}^{\pi_n} \right)$$

$$\overset{(i)}{\leq} \sum_{k \leq K} \sum_{n \leq N} \max_{\hat{\theta}_n^k, \tilde{\theta}_n^k \in \boldsymbol{\mathcal{B}}_k} \mathsf{D}_{\mathsf{TV}} \left( \mathbb{P}_{\hat{\theta}_n^k}^{\pi^{n,k}}, \mathbb{P}_{\tilde{\theta}_n^k}^{\pi^{n,k}} \right)$$

$$\leq 2 \sum_k \sum_n \max_{\hat{\theta}_n^k \in \boldsymbol{\mathcal{B}}_k} \mathsf{D}_{\mathsf{TV}} \left( \mathbb{P}_{\hat{\theta}_n^k}^{\pi^{n,k}}, \mathbb{P}_{\theta_n^*}^{\pi^{n,k}} \right)$$

$$\leq \sum_{k \leq K} \frac{8Q_A\sqrt{|\mathcal{A}|\beta^{(N)}}}{\gamma} \sqrt{\sum_{n \leq N} \sum_{h \leq H} \mathbb{E}_{\tau_h \sim \mathbb{P}_{\theta_n^*}^{\pi^{n,k}}} \left[ \left\| (\mathbf{A}_h^n)^\dagger \bar{\psi}^{n,*}(\tau_{h-1}) \right\|_{(U_{k,h}^n)^{-1}}^2 \right]}$$

$$\overset{(ii)}{\leq} \frac{8Q_A\sqrt{K|\mathcal{A}|\beta^{(N)}}}{\gamma} \sqrt{\sum_{k \leq K} \sum_{n \leq N} \sum_{h \leq H} \mathbb{E}_{\tau_h \sim \mathbb{P}_{\theta_n^*}^{\pi^{n,k}}} \left[ \left\| (\mathbf{A}_h^n)^\dagger \bar{\psi}^{n,*}(\tau_{h-1}) \right\|_{(U_{k,h}^n)^{-1}}^2 \right]}$$

$$\overset{(iii)}{\leq} \frac{8Q_A\sqrt{K|\mathcal{A}|\beta}}{\gamma} \sqrt{rNH\log(1 + rK/\lambda)},$$

where $(i)$ follows from the fact that $\bar{\theta}_n \in \boldsymbol{\mathcal{B}}_k$ for all $k \in [K]$, $(ii)$ is due to the Cauchy's inequality, and $(iii)$ follows from Lemma 9.

Hence,

$$\sum_n \mathsf{D}_{\mathsf{TV}} \left( \mathbb{P}_{\bar{\theta}_n}^{\pi_n}, \mathbb{P}_{\theta_n^*}^{\pi_n} \right) \leq \tilde{O} \left( \frac{Q_A\sqrt{r|\mathcal{A}|NH\beta^{(N)}\log(1 + rK/\lambda)}}{\gamma\sqrt{K}} \right).$$

$\square$

## D  DOWNSTREAM LEARNING: PROOF OF THEOREM 2

In this section, we first provide the full algorithm of OMLE. Then, we prove a new estimation guarantee under the presence of approximation error of $\hat{\Theta}_0^{\mathsf{u}}$. Finally, we provide the proof of Theorem 2.

### D.1 Optimistic model-based algorithm

In this section, we provide the full algorithm of OMLE (Liu et al., 2022; Chen et al., 2022) given a model class $\hat{\Theta}$ and an estimation margin $\beta_0$, for the completeness of the paper.

First, OMLE seeks a exploration policy $\pi^k$ that maximizes the largest total variation distance between any two model parameters $\theta$ and $\theta'$ within a confidence set $\mathcal{B}_k$. Then, OMLE uses policies adapted from $\pi^k$ to collect data. Finally, using the collected sample trajectories, OMLE constructs a confidence set which includes the true model parameter. The pseudo code is provided in Algorithm 2.

---

**Algorithm 2** Downstream multi-task PSR (OMLE)

---

1: **Input:** $\mathcal{B}_1 = \hat{\Theta}_0^{\mathrm{u}}$, estimation margin $\beta_0$.
2: **for** $k = 1, \ldots, K_{\mathrm{Down}}$ **do**
3:
$$\pi^k = \arg\max_{\pi \in \Pi} \max_{\theta, \theta' \in \mathcal{B}_k} \mathrm{D}_{\mathrm{TV}}\left(\mathbb{P}_\theta^\pi, \mathbb{P}_{\theta'}^\pi\right)$$
4:     **for** $h \in [H]$ **do**
5:         Use $\nu_h^{\pi^k}$ to collect data $\tau_H^{k,h}$.
6:     **end for**
7:     Construct confidence set $\mathcal{B}_{k+1}$ :

$$\mathcal{B}_{k+1} = \left\{ \theta \in \hat{\Theta} : \sum_{t<k}\sum_h \log \mathbb{P}_\theta^{\nu_h^{\pi^t}}\left(\tau_H^{t,h}\right) \geq \max_{\theta' \in \hat{\Theta}} \sum_{t<k}\sum_h \log \mathbb{P}_\theta^{\nu_h^{\pi^t}}\left(\tau_H^{t,h}\right) - \beta_0 \right\} \cap \mathcal{B}_k$$

8: **end for**
9: **Output:** Any $\bar{\theta}_0 \in \mathcal{B}_{K_{\mathrm{Down}}+1}$, and a greedy policy $\bar{\pi}_0 = \arg\max_\pi V_{\bar{\theta}_0, R_0}^\pi$.

---

### D.2 Estimation Guarantee of OMLE

Recall that $\epsilon_0 = \mathrm{e}_\alpha(\hat{\Theta}_0^{\mathrm{u}}) = \min_{\theta_0 \in \hat{\Theta}_0^{\mathrm{u}}} \max_\pi \mathrm{D}_{\mathrm{R},\alpha}\left(\mathbb{P}_{\theta_0^*}^\pi, \mathbb{P}_{\theta_0}^\pi\right)$ is the approximation error of the model class $\hat{\Theta}_0^{\mathrm{u}}$. In this section, let $\theta_0^{\epsilon_0} = \arg\min_{\theta_0 \in \hat{\Theta}_0^{\mathrm{u}}} \max_\pi \mathrm{D}_{\mathrm{R},\alpha}\left(\mathbb{P}_{\theta_0^*}^\pi, \mathbb{P}_{\theta_0}^\pi\right)$.

The following lemma is from Proposition B.1 in Liu et al. (2022).

**Lemma 3.** *Let* $\eta \leq \frac{1}{KH}$. *With probability at least* $1 - \delta$, *for any* $\theta_0 \in \hat{\Theta}_0^{\mathrm{u}}$, *we have*

$$\sum_{t<k}\sum_h \log \frac{\mathbb{P}_{\theta_0}^{\nu_h^{\pi^t}}\left(\tau_H^{t,h}\right)}{\mathbb{P}_{\theta_0^*}^{\nu_h^{\pi^t}}\left(\tau_H^{t,h}\right)} \leq \log(\mathcal{N}_\eta(\hat{\Theta}_0^{\mathrm{u}})) + \log \frac{eK}{\delta}.$$

Then, we show that the log-likelihood of model $\theta_0^{\epsilon_0}$ is sufficiently large.

**Lemma 4.** *With probability at least* $1 - \delta$, *we have*

$$\sum_{t<k}\sum_h \log \frac{\mathbb{P}_{\theta_0^*}^{\nu_h^{\pi^t}}\left(\tau_H^{t,h}\right)}{\mathbb{P}_{\theta_0^{\epsilon_0}}^{\nu_h^{\pi^t}}\left(\tau_H^{t,h}\right)} \leq \epsilon_0 KH + \frac{\mathbf{1}_{\{\epsilon_0 \neq 0\}}}{\alpha - 1} \log \frac{K}{\delta}.$$

*Proof.* By the definition of $\theta_0^{\epsilon_0}$, we have

$$\frac{1}{\alpha - 1} \log \mathbb{E}_{\mathbb{P}_{\theta^*}^\pi} \left[ \left( \frac{\mathbb{P}_{\theta^*}^{\pi'}(\tau_H)}{\mathbb{P}_{\theta^{\epsilon_0}}^{\pi'}(\tau_H)} \right)^{\alpha - 1} \right] \leq \epsilon_0. \tag{12}$$

If $\epsilon_0 = 0$, then the proof is trivial. In the following, we mainly consider the case when $\epsilon_0 > 0$.

By the Markov's inequality, for any $x \in \mathbb{R}$, we have

$$
\mathbb{P}\left(\sum_{t<k}\sum_h \log \frac{\mathbb{P}_{\theta^*}^{\pi^{t,h}}(\tau_H^{t,h})}{\mathbb{P}_{\theta^{\epsilon_0}}^{\pi^{t,h}}(\tau_H^{t,h})} \geq x\right)
$$

$$
= \mathbb{P}\left(\prod_{t<k}\prod_h \left(\frac{\mathbb{P}_{\theta^*}^{\pi^{t,h}}(\tau_H^{t,h})}{\mathbb{P}_{\theta^{\epsilon_0}}^{\pi^{t,h}}(\tau_H^{t,h})}\right)^{\alpha-1} \geq e^{(\alpha-1)x}\right)
$$

$$
\leq e^{-(\alpha-1)x}\mathbb{E}\left[\prod_{t<k}\prod_h \left(\frac{\mathbb{P}_{\theta^*}^{\pi^{t,h}}(\tau_H^{t,h})}{\mathbb{P}_{\theta^{\epsilon_0}}^{\pi^{t,h}}(\tau_H^{t,h})}\right)^{\alpha-1} \mathbb{E}\left[\left(\frac{\mathbb{P}_{\theta^*}^{\pi^{k,h}}(\tau_H^{k,h})}{\mathbb{P}_{\theta^{\epsilon_0}}^{\pi^{k,h}}(\tau_H^{k,h})}\right)^{\alpha-1}\middle| \pi^{k,h}\right]\right]
$$

$$
\overset{(i)}{\leq} e^{-(\alpha-1)x}e^{(\alpha-1)KH\epsilon_0}
$$

$$
= e^{-(\alpha-1)(x-KH\epsilon_0)},
$$

where $(i)$ follows from Equation (12).

By choosing $x = \epsilon_0 KH + \frac{1}{\alpha-1}\log(K/\delta)$ and taking union bound over $k$, we conclude that, with probability at least $1 - \delta$,

$$
\sum_{t<k}\sum_h \log \frac{\mathbb{P}_{\theta^*}^{\pi^{t,h}}(\tau_H^{t,h})}{\mathbb{P}_{\theta^{\epsilon_0}}^{\pi^{t,h}}(\tau_H^{t,h})} \leq \epsilon_0 KH + \frac{1}{\alpha-1}\log\frac{K}{\delta}.
$$

$\square$

Combining Lemma 3 and Lemma 4, we immediately obtain that with probability at least $1 - \delta/2$, the following bound holds.

$$
\sum_{t<k}\sum_h \log \mathbb{P}_{\theta_0^{\epsilon_0}}^{\nu_h^{\pi^t}}(\tau_H^{t,h}) \geq \max_{\theta_0\in\hat{\Theta}_0^u}\sum_{t<k}\sum_h \log \mathbb{P}_{\theta_0}^{\nu_h^{\pi^t}}(\tau_H^{t,h})
$$
$$
- \left(\log\mathcal{N}_\eta(\hat{\Theta}_0^u) + \log\frac{4eK}{\delta} + \epsilon_0 KH + \frac{\mathbf{1}_{\{\epsilon_0\neq 0\}}}{\alpha-1}\log\frac{4K}{\delta}\right),
$$

where $\eta \leq \frac{1}{KH}$.

Setting $\beta_0 = \log\mathcal{N}_\eta(\hat{\Theta}_0^u) + \log\frac{4eK}{\delta} + \epsilon_0 KH + \frac{\mathbf{1}_{\{\epsilon_0\neq 0\}}}{\alpha-1}\log\frac{4K}{\delta}$, we conclude that $\theta_0^{\epsilon_0} \in \mathcal{B}_k$ for all $k \in [K]$. Based on this fact, we have the following estimation guarantee.

**Lemma 5.** *With probability at least $1 - \delta$, for any $k \in [K]$ and $\theta_0 \in \mathcal{B}_k$, we have*

$$
\sum_{t<k}\sum_h \mathrm{D}_{\mathrm{TV}}^2\left(\mathbb{P}_{\theta_0}^{\nu_h^{\pi^t}}, \mathbb{P}_{\theta_0^*}^{\nu_h^{\pi^t}}\right) \leq 2\beta_0.
$$

*Proof.* We follow the same argument as in Proposition 2, except setting $N = 1$ and $\Theta_u = \hat{\Theta}_0^u$. Then, we obtain that, with probability at least $1 - \delta/2$, the following inequality holds.

$$
\sum_{t<k}\sum_h \mathrm{D}_{\mathrm{H}}^2\left(\mathbb{P}_{\theta_0}^{\nu_h^{\pi^t}}, \mathbb{P}_{\theta_0^*}^{\nu_h^{\pi^t}}\right) \leq \sum_{t\leq k}\sum_h \log \frac{\mathbb{P}_{\theta_0^*}^{\nu_h^{\pi^t}}(\tau_H^{t,h})}{\mathbb{P}_{\theta_0}^{\nu_h^{\pi^t}}(\tau_H^{t,h})} + \log\frac{K\mathcal{N}_\eta(\hat{\Theta}_0^u)}{\delta}.
$$

Since $\theta_0, \theta_0^{\epsilon_0} \in \mathcal{B}_k$, by the optimality of $\mathcal{B}_k$, we further have

$$
\sum_{t<k}\sum_h \mathrm{D}_{\mathrm{H}}^2\left(\mathbb{P}_{\theta_0}^{\nu_h^{\pi^t}}, \mathbb{P}_{\theta_0^*}^{\nu_h^{\pi^t}}\right)
$$

$$\leq \sum_{t \leq k} \sum_h \log \frac{\mathbb{P}_{\theta_0^*}^{\nu_h^{\pi^t}}(\tau_H^{t,h})}{\mathbb{P}_{\theta_0^{\epsilon_0}}^{\nu_h^{\pi^t}}(\tau_H^{t,h})} + \beta_0 + \log \frac{K\mathcal{N}_\eta(\hat{\Theta}_0^{\mathrm{u}})}{\delta}$$

$$\overset{(i)}{\leq} \epsilon_0 KH + \frac{\mathbf{1}_{\{\epsilon_0 \neq 0\}}}{\alpha - 1} \log \frac{K}{\delta} + \beta_0 + \log \frac{K\mathcal{N}_\eta(\hat{\Theta}_0^{\mathrm{u}})}{\delta}$$

$$\leq 2\beta_0,$$

where $(i)$ is due to Lemma 4.

$\square$

### D.3 PROOF OF THEOREM 2

**Theorem 4** (Restatement of Theorem 2). *Fix $\alpha > 1$. Let $\epsilon_0 = \mathrm{e}_\alpha(\hat{\Theta}_0^{\mathrm{u}})$, $\beta_0 = O(\log \frac{KH}{\delta} + \log \mathcal{N}_\eta(\hat{\Theta}_0^{\mathrm{u}}) + \epsilon_0 KH + \frac{\mathbf{1}_{\{\epsilon_0 \neq 0\}}}{\alpha - 1})$, where $\eta \leq \frac{1}{KH}$. Under Assumption 1, with probability at least $1 - \delta$, the output of Algorithm 2 satisfies that*

$$\max_{\pi \in \Pi} \mathsf{D}_{\mathrm{TV}}\left(\mathbb{P}_{\bar{\theta}_0}^\pi, \mathbb{P}_{\theta_0^*}^\pi\right) \leq \tilde{O}\left(\frac{Q_A}{\gamma}\sqrt{\frac{r|\mathcal{A}|H\beta_0}{K}} + \sqrt{\epsilon_0}\right). \tag{13}$$

*Proof.* First, we follow the proof in Lemma 2, except setting $N = 1$. We obtain that

$$\mathsf{D}_{\mathrm{TV}}\left(\mathbb{P}_{\hat{\theta}_0^k}^\pi(\tau_H), \mathbb{P}_{\theta_0^*}^\pi(\tau_H)\right) \leq \frac{Q_A|\mathcal{A}|}{\gamma}\sqrt{C_\lambda + \sum_{t<k}\zeta_{t,h}^0} \sum_h \mathbb{E}_{\tau_{h-1}\sim\mathbb{P}_{\theta_0^*}^\pi}\left[\|(\mathbf{A}_h^0)^\dagger \bar{\psi}^{0,*}(\tau_{h-1})\|_{(U_{k,h}^0)^{-1}}\right]$$

$$\overset{(i)}{\leq} O\left(\frac{Q_A\sqrt{|\mathcal{A}|\beta_0}}{\gamma} \sum_h \mathbb{E}_{\tau_{h-1}\sim\mathbb{P}_{\theta_0^*}^\pi}\left[\|(\mathbf{A}_h^0)^\dagger \bar{\psi}^{0,*}(\tau_{h-1})\|_{(U_{k,h}^0)^{-1}}\right]\right),$$

where

$$\begin{cases} C_\lambda = \dfrac{\lambda r Q_A^2 |\mathcal{A}|}{\gamma^4}, \\[2mm] \lambda = \dfrac{\gamma^4 \beta_0}{r Q_A^2 |\mathcal{A}|}, \\[2mm] \zeta_{t,h}^0 = \mathsf{D}_{\mathrm{H}}^2\left(\mathbb{P}_{\hat{\theta}_0^k}^{\nu_h^{\pi^t}}(\tau_H^{t,h}), \mathbb{P}_{\theta_0^*}^{\nu_h^{\pi^t}}(\tau_H^{t,h})\right), \\[2mm] U_{k,h}^0 = \lambda I + (\mathbf{A}_h^0)^\dagger \displaystyle\sum_{t<k} \mathbb{E}_{\tau_{h-1}\sim\mathbb{P}_{\theta_0^*}}^{\nu_h^{\pi^t}} \bar{\psi}^{0,*}(\tau_{h-1})\bar{\psi}^{0,*}(\tau_{h-1})^\top ((\mathbf{A}_h^0)^\dagger)^\top, \end{cases} \tag{14}$$

and $(i)$ is due to the estimation guarantee.

Therefore,

$$K\mathsf{D}_{\mathrm{TV}}\left(\mathbb{P}_{\bar{\theta}_0}^\pi, \mathbb{P}_{\theta_0^*}^\pi\right)$$

$$\leq K\mathsf{D}_{\mathrm{TV}}\left(\mathbb{P}_{\bar{\theta}_0}^\pi, \mathbb{P}_{\theta_0^{\epsilon_0}}^\pi\right) + K\mathsf{D}_{\mathrm{TV}}\left(\mathbb{P}_{\theta_0^{\epsilon_0}}^\pi, \mathbb{P}_{\theta_0^*}^\pi\right)$$

$$\leq \sum_k \max_{\hat{\theta}_0^k, \tilde{\theta}_0^k \in \mathcal{B}_k} \mathsf{D}_{\mathrm{TV}}\left(\mathbb{P}_{\hat{\theta}_0^k}^{\pi^k}, \mathbb{P}_{\tilde{\theta}_0^k}^{\pi^k}\right) + K\sqrt{\epsilon_0}$$

$$\leq 2\sum_k \max_{\hat{\theta}_0^k \in \mathcal{B}_k} \mathsf{D}_{\mathrm{TV}}\left(\mathbb{P}_{\hat{\theta}_0^k}^{\pi^k}, \mathbb{P}_{\theta_0^*}^{\pi^k}\right) + K\sqrt{\epsilon_0}$$

$$\leq \frac{Q_A\sqrt{|\mathcal{A}|(\beta_0 + \epsilon_0 KH + \frac{1}{\alpha-1}\log(K/\delta))}}{\gamma}\sqrt{rHK\log(1 + rK/\lambda)} + K\sqrt{\epsilon_0}.$$

$\square$

# E    BRACKETING NUMBERS OF EXAMPLES AND MISSING PROOFS IN SECTION 4.3 AND SECTION 5.2

In the section, we present bracketing numbers of examples and missing Proofs in Section 4.3 and Section 5.2. Because $\phi_h^\top = \sum_{(o_h,a_h)\in\mathcal{O}\times\mathcal{A}} \phi_{h+1}^\top \mathbf{M}_h(o_h, a_h)$ for each $h \in [H-1]$, $\phi_h$ can be decided by $\phi_H$ and $\{\mathbf{M}_h\}_{h=1}^H$. For simplicity, in this section, we reparameterize the PSRs parameters as $\theta = \{\phi_H, \{\mathbf{M}_h\}_{h=1}^H\}$. Without loss of generality, from Theorem C.1 and C.7 in Liu et al. (2022), we assume for any $(o, a) \in \mathcal{O} \times \mathcal{A}$, $\mathbf{M}_h(o, a) \in \mathbb{R}^{r \times r}$, and the rank of $\mathbf{M}_h(o, a)$ is $r$.

## E.1    BRACKETING NUMBERS OF BASIC SINGLE-TASK PSRs

For completeness, in this subsection, we first present the analysis of bracketing number of basic single-task PSRs.

**Lemma 6** (Bracketing number of single-task PSRs). *Let $\Theta$ be the collection of PSR parameters of all rank-$r$ sequential decision making problems with obseration space $\mathcal{O}$, action space $\mathcal{A}$ and horizon $H$. Then we have*

$$\log \mathcal{N}_\eta(\Theta) \le O(r^2 H^2 |\mathcal{O}||\mathcal{A}| \log(\frac{|\mathcal{O}||\mathcal{A}|}{\eta})).$$

*Proof.* We assume $\psi_0$ is known,[5] and $\|\psi_0\|_2 \le \sqrt{|\mathcal{A}|^H}$. By Corollary C.8 from Liu et al. (2022), the PSR model class have following form:

$$\Theta = \left\{ \theta : \theta = \{\phi_H, \{\mathbf{M}_h\}_{h=1}^H\}, \|\mathbf{M}_h(o, a)\|_2 \le 1 \, \text{for} \, (o, a) \in \mathcal{O} \times \mathcal{A}, \|\phi_H\|_2 \le 1 \right\}.$$

Denote $\widetilde{\Theta}_\delta$ as the $\delta$-cover of $\Theta$ w.r.t $\ell_\infty$-norm with $\delta = \frac{\eta}{(|\mathcal{O}||\mathcal{A}|)^{cH}}$ for some large $c > 0$. Mathematically, for any $\theta = \{\phi_H, \{\mathbf{M}_h\}_{h=1}^H\} \in \Theta$, there exists $\tilde{\theta} = \{\tilde{\phi}_H, \{\widetilde{\mathbf{M}}_h\}_{h=1}^H\} \in \widetilde{\Theta}_\delta$, such that for any $(o, a) \in \mathcal{O} \times \mathcal{A}$,

$$\left\| \phi_H - \tilde{\phi}_H \right\|_\infty \le \delta, \quad \left\| \mathbf{Vec}(\mathbf{M}_h(o, a)) - \mathbf{Vec}(\widetilde{\mathbf{M}}_h(o, a)) \right\|_\infty \le \delta.$$

Next, we show that $\widetilde{\Theta}_\delta$ can constitute an $\eta$-bracket for $\Theta$. For any policy $\pi$, we have

$$\sum_{\tau_H \in (\mathcal{O}\times\mathcal{A})^H} \left| \mathbb{P}_\theta(\tau_H) - \mathbb{P}_{\tilde{\theta}}(\tau_H) \right| \times \pi(\tau_H)$$

$$\le \sum_{\tau_H \in (\mathcal{O}\times\mathcal{A})^H} \sum_{h=1}^H \left| \mathbf{m}_h(\omega_h)^\top \left( \widetilde{\mathbf{M}}_h(o_h, a_h) - \mathbf{M}_h(o_h, a_h) \right) \psi_{h-1}(\tau_{h-1}) \right| \times \pi(\tau_H)$$

$$\overset{(i)}{\le} \sum_{\tau_H \in (\mathcal{O}\times\mathcal{A})^H} \sum_{h=1}^H \|\mathbf{m}_h(\omega_h)\|_2 \left\| \left( \widetilde{\mathbf{M}}_h(o_h, a_h) - \mathbf{M}_h(o_h, a_h) \right) \right\|_2 \|\psi_{h-1}(\tau_{h-1})\|_2 \times \pi(\tau_H)$$

$$\overset{(ii)}{\le} \sum_{\tau_H \in (\mathcal{O}\times\mathcal{A})^H} \sum_{h=1}^H \left\| \left( \widetilde{\mathbf{M}}_h(o_h, a_h) - \mathbf{M}_h(o_h, a_h) \right) \right\|_2 \sqrt{|\mathcal{A}|^H} \times \pi(\tau_H)$$

$$\overset{(iii)}{\le} \sum_{\tau_H \in (\mathcal{O}\times\mathcal{A})^H} \sum_{h=1}^H \sqrt{r} \left\| \left( \widetilde{\mathbf{M}}_h(o_h, a_h) - \mathbf{M}_h(o_h, a_h) \right) \right\|_\infty \sqrt{|\mathcal{A}|^H} \times \pi(\tau_H) \qquad (15)$$

$$\le \sum_{\tau_H \in (\mathcal{O}\times\mathcal{A})^H} \sum_{h=1}^H \sqrt{r^3} \left\| \left( \mathbf{Vec}(\widetilde{\mathbf{M}}_h(o_h, a_h)) - \mathbf{Vec}(\mathbf{M}_h(o_h, a_h)) \right) \right\|_\infty \sqrt{|\mathcal{A}|^H} \times \pi(\tau_H)$$

$$\le H\sqrt{r^3 |\mathcal{A}|^H} \delta \le \eta,$$

---

[5]Such assumption does not influence the order of bracketing number since the model complexity related to $\Psi_0$ does not dominate.

where $(i)$ follows from the property of operation norm of matrix, $(ii)$ follows from the fact that $\|\mathbf{m}_h(\omega_h)\|_2 \leq \|\phi_H\|_2 \|\mathbf{M}_H(o_H, a_H)\|_2 \ldots \|\mathbf{M}_{h+1}(o_{h+1}, a_{h+1})\|_2 \leq 1$ and $\|\psi_{h-1}(\tau_{h-1})\| \leq \|\mathbf{M}_{h-1}(o_{h-1}, a_{h-1})\|_2 \ldots \|\mathbf{M}_1(o_1, a_1)\|_2 \|\psi_0\|_2 \leq \sqrt{|\mathcal{A}|^H}$, and $(iii)$ follows from the relationship between the operation norm induced by $\ell_2$ norm and $\ell_\infty$ norm.

By Lemma 10,

$$|\widetilde{\Theta}_\delta| \leq \left(\frac{1 + 2\sqrt{r}}{\delta}\right)^{r + r^2 \times H|\mathcal{O}||\mathcal{A}|} = \left(3\sqrt{r}\frac{(|\mathcal{O}||\mathcal{A}|)^{cH}}{\eta}\right)^{r + r^2 \times H|\mathcal{O}||\mathcal{A}|}$$

and

$$\log |\widetilde{\Theta}_\delta| = O(r^2 H^2 |\mathcal{O}||\mathcal{A}| \log(\frac{|\mathcal{O}||\mathcal{A}|}{\eta})),$$

which equals to the log $\eta$-bracketing number.

$\square$

## E.2 Bracketing Number of Upstream examples

**Example 1(Multi-task POMDP with same transition kernels):** The multi-task parameter space is

$$\left\{(\mathbb{T}_{h,a}, \mathbb{O}_h^1, \ldots, \mathbb{O}_h^N) : \mathbb{T}_{h,a} \in \mathbb{R}^{|\mathcal{S}| \times |\mathcal{S}|}, \mathbb{O}_h^i \in \mathbb{R}^{|\mathcal{O}| \times |\mathcal{A}|}, \forall i \in [N]\right\}_{h \in [H], a \in \mathcal{A}}.$$

Note that the value of each coordinate of these matrices are probabilities, and thus are bounded within $[0,1]$. Therefore, the $\eta$-bracketing number in this case is $O(H(|\mathcal{S}|^2|\mathcal{A}| + N|\mathcal{O}||\mathcal{S}|) \log \frac{H|\mathcal{S}||\mathcal{O}||\mathcal{A}|}{\eta})$

**Example 2(Multi-task PSR with perturbed models):** Suppose there exist a latent base task $P_b$, and a noisy perturbation space $\boldsymbol{\Delta}$. Each task $n \in [N]$ is a noisy perturbation of the latent base task and can be parameterized into two parts: the base task plus a task-specified noise term. Specifically, for each step $h \in [H]$ and task $n \in [N]$, any $(o, a) \in \mathcal{O} \times \mathcal{A}$, we have

$$\mathbf{M}_h^n(o_h, a_h) = \mathbf{M}_h^b(o_h, a_h) + \Delta_h^n(o_h, a_h), \quad \Delta_h^n \in \boldsymbol{\Delta}.$$

Such a multi-task PSR satisfies that $\beta^{(N)} \leq O(\log \frac{KHN}{\delta} + r^2|\mathcal{O}||\mathcal{A}|H^2 \log \frac{|\mathcal{O}||\mathcal{A}|}{\eta} + HN \log |\boldsymbol{\Delta}|)$, whereas $\beta^{(1)}$ for a single task is given by $O(r^2|\mathcal{O}||\mathcal{A}|H^2 \log \frac{|\mathcal{O}||\mathcal{A}|}{\eta} + H(N-1) \log |\boldsymbol{\Delta}|)$. Clearly, $\beta^{(N)} \ll N\beta^{(1)}$ holds if $\log |\boldsymbol{\Delta}| \ll \tilde{O}(r^2|\mathcal{O}||\mathcal{A}|H)$, which can be easily satisfied for low perturbation environments. In such a case, the multi-task PSR benefits from a significantly reduced sample complexity compared to single-task learning.

*Proof of Example 2.* Suppose there exist a latent base task model space:

$$\Theta^b = \left\{\theta : \theta = \{\phi_H, \{\mathbf{M}_h\}_{h=1}^H\}, \|\mathbf{M}_h(o, a)\|_2 \leq 1 \text{ for } (o, a) \in \mathcal{O} \times \mathcal{A}, \|\phi_H\|_2 \leq 1\right\}.$$

A base task model is selected: $\theta^b = \{\phi_H^b, \{\mathbf{M}_h^b\}_{h=1}^H\}$. The parameters of each task $n$ in multi-task PSR models are as follows:

$$\Theta^n = \left\{\theta : \theta = \{\phi_H^b, \{\mathbf{M}_h^b + \Delta_h^n\}_{h=1}^H\}, \Delta_h^n \in \boldsymbol{\Delta}, \|\phi_H^n\|_2 \leq 1\right\},$$

where $\Delta_h^n(\cdot, \cdot) : \mathcal{O} \times \mathcal{A} \to \mathbb{R}^{d_h \times d_{h-1}}$ for any $h \in [H]$, and $n \in [N]$, and $\boldsymbol{\Delta}$ is the noisy perturbation space with finite cardinality.

Let $\widetilde{\Theta}_\delta^b$ be the $\delta$-cover of $\Theta^b$ w.r.t $\ell_\infty$-norm with $\delta = \frac{\eta}{(|\mathcal{O}||\mathcal{A}|)^{cH}}$. From Lemma 6, we have $|\widetilde{\Theta}_\delta^b| = \left(\frac{(|\mathcal{O}||\mathcal{A}|)^{cH}}{\eta}\right)^{2r + r^2 \times H|\mathcal{O}||\mathcal{A}|}$. For each $n \in [N]$, denote

$$\widetilde{\Theta}_\delta^n = \widetilde{\Theta}_\delta^b + \boldsymbol{\Delta}$$
$$:= \left\{\theta : \theta = \left(\widetilde{\phi}_H^b, \{\widetilde{\mathbf{M}}_h^n + \Delta_h^n\}_{h=1}^H\right); \left(\widetilde{\phi}_H^b, \{\widetilde{\mathbf{M}}_h\}_{h=1}^H\right) \in \widetilde{\Theta}_\delta^b; \Delta_h^n \in \boldsymbol{\Delta}\right\}.$$

Obviously, $|\widetilde{\Theta}_\delta^n| = \left(3\sqrt{r}\frac{(|\mathcal{O}||\mathcal{A}|)^{cH}}{\eta}\right)^{r+r^2 \times H|\mathcal{O}||\mathcal{A}|} \times |\mathbf{\Delta}|^H$. Then denote the multi-task $\delta$-cover as $\widetilde{\mathbf{\Theta}}_\delta = \widetilde{\Theta}_\delta^1 \times \cdots \times \widetilde{\Theta}_\delta^N$. Following from Lemma 6, for any task $n \geq 1$, $\widetilde{\Theta}_\delta^n$ can constitute an $\eta$-bracket for $\Theta^n$, so $\widetilde{\mathbf{\Theta}}_\delta$ can constitute an $\eta$-bracket for $\mathbf{\Theta}$. By noticing that each $\bar{\Theta}_\delta^n$ has a common part, and the changing part is only related to $\Delta_h^n$. The corresponding $\eta$-bracketing number is $\left(3\sqrt{r}\frac{(|\mathcal{O}||\mathcal{A}|)^{cH}}{\eta}\right)^{r+r^2 \times H|\mathcal{O}||\mathcal{A}|} \times |\mathbf{\Delta}|^{HN}$, and the log $\eta$-bracketing number is at most $O\left(r^2|\mathcal{O}||\mathcal{A}|H^2 \log\frac{|\mathcal{O}||\mathcal{A}|}{\eta} + H(N-1)\log|\mathbf{\Delta}|\right)$. $\qquad\square$

**Example 3(Multi-task PSRs: Linear combination of core tasks):** Suppose that the multi-task PSR lies in the linear span of $m$ core tasks, i.e., there exist a set of core tasks indexed by $\{1, 2, \ldots, m\}$ such that each PSR can be represented as a linear combination of those $m$ core tasks. Specifically, for each task $n \in [N]$, there exists a coefficient vector $\boldsymbol{\alpha}^n = (\alpha_1^n, \cdots, \alpha_m^n)^\top \in \mathbb{R}^m$ s.t. for any $h \in [H]$ and $(o_h, a_h) \in \mathcal{O} \times \mathcal{A}$,

$$\phi_h^n(o_h, a_h) = \sum_{l=1}^m \alpha_l^n \phi_h^l(o_h, a_h), \quad \mathbf{M}_h^n(o_h, a_h) = \sum_{l=1}^m \alpha_l^n \mathbf{M}_h^l(o_h, a_h).$$

For regularization, we assume $0 \leq \alpha_l^n \leq 1$ for all $l \in [m]$ and $n \in [N]$, and $\sum_{l=1}^m \alpha_l^n = 1$ for all $n \in [N]$. It can be shown that $\beta^{(N)} = O(m(r^2|\mathcal{O}||\mathcal{A}|H^2 + N)\log\frac{|\mathcal{O}||\mathcal{A}|}{\eta})$, whereas $\beta^{(1)}$ for a single task is given by $r^2|\mathcal{O}||\mathcal{A}|H^2 \log\frac{|\mathcal{O}||\mathcal{A}|}{\eta}$. Clearly, $\beta^{(N)} \ll N\beta^{(1)}$ holds if $m \leq \min\{N, r^2|\mathcal{O}||\mathcal{A}|H^2\}$, which is satisfied in practice.

*Proof of Example 3.* Denote the core tasks model class $\mathbf{\Theta}^0 = \Theta^{0,1} \times \cdots \times \Theta^{0,m}$, and the multi-task model class $\mathbf{\Theta} = \Theta^1 \times \cdots \times \Theta^N$, where for any $n \in [N]$, $\Theta^n$ is defined as:

$$\Theta^n = \left\{ \left( \sum_{l=1}^m \alpha_l^n \phi_H^l, \left\{ \sum_{l=1}^m \alpha_l^n \mathbf{M}_h^l \right\}_{h=1}^H \right) : \boldsymbol{\alpha}^n = (\alpha_1^n, \cdots, \alpha_m^n)^\top \in \mathbb{R}^m; \left( \phi_H^l, \{\mathbf{M}_h^l\}_{h=1}^H \right) \in \Theta^{0,l} \right\}.$$

For any $\delta \geq 0$, we first consider the $\delta$-cover of the model class of core tasks $\widetilde{\mathbf{\Theta}}_\delta^0 = \widetilde{\Theta}_\delta^{0,1} \times \cdots \times \widetilde{\Theta}_\delta^{0,m}$, where for each base tasks $l \in [m]$, $\widetilde{\Theta}_\delta^l$ is a $\delta$-cover for $\Theta^m$. Similar to proof of Lemma 6, $|\widetilde{\Theta}_\delta^{0,l}| = (3\sqrt{r} \times \frac{1}{\delta})^{r^2 H^2 |\mathcal{O}||\mathcal{A}|}$ and $|\widetilde{\mathbf{\Theta}}_\delta^0| = (3\sqrt{r} \times \frac{1}{\delta})^{r^2 H^2 |\mathcal{O}||\mathcal{A}|m}$.

Then we consider the cover for multi-task parameter space. Denote $C_\delta^m$ as a $\delta$-cover for the unit ball in $\mathbb{R}^m$ w.r.t. $\ell_1$ norm. Mathematically, for any $n \in [N]$ and vector $\boldsymbol{\alpha}^n \in \mathbb{R}^m$, there exists an $\tilde{\boldsymbol{\alpha}}^n \in C_\delta^m$ such that $\|\boldsymbol{\alpha}^n - \tilde{\boldsymbol{\alpha}}^n\|_1 \leq \delta$. In addition, the cardinality $|C_\delta^m| = (\frac{3}{\delta})^m$.

Define the multi-task model class $\widetilde{\mathbf{\Theta}}_\delta = \widetilde{\Theta}_\delta^1 \times \cdots \times \widetilde{\Theta}_\delta^N$, where for any $n \in [N]$, $\widetilde{\Theta}_\delta^n$ is defined as

$$\widetilde{\Theta}_\delta^n = \left\{ \left( \sum_{l=1}^m \tilde{\alpha}_l^n \widetilde{\phi}_H^l, \left\{ \sum_{l=1}^m \tilde{\alpha}_l^n \widetilde{\mathbf{M}}_h^l \right\}_{h=1}^H \right) : \tilde{\boldsymbol{\alpha}}^n = (\tilde{\alpha}_1^n, \cdots, \tilde{\alpha}_m^n)^\top \in C_\delta^m; \left( \widetilde{\phi}_H^l, \{\widetilde{\mathbf{M}}_h^l\}_{h=1}^H \right) \in \widetilde{\Theta}_\delta^{0,l} \right\}.$$

We next show that $\widetilde{\mathbf{\Theta}}_\delta$ is a $\eta$-bracket of $\mathbf{\Theta}_\delta$.

By definition, for any model of base task $l \in [m]$: $\theta^l = \left( \phi_H^l, \{\mathbf{M}_h^l\}_{h=1}^H \right) \in \Theta^l$, there exists $\tilde{\theta}^l = \left( \widetilde{\phi}_H^l, \{\widetilde{\mathbf{M}}_h^l\}_{h=1}^H \right) \in \widetilde{\Theta}_\delta^l$, such that for any $(o, a) \in \mathcal{O} \times \mathcal{A}$,

$$\left\| \phi_H^l - \tilde{\phi}_H^l \right\|_\infty \leq \delta, \left\| \mathbf{Vec}(\mathbf{M}_h^l(o, a)) - \mathbf{Vec}(\widetilde{\mathbf{M}}_h^l(o, a)) \right\|_\infty \leq \delta.$$

Then, for any $\theta^l \in \Theta^l$, $\boldsymbol{\alpha}^n \in \mathbb{R}^m$, there exist $\tilde{\theta} \in \tilde{\Theta}_\delta^l$ and $\tilde{\boldsymbol{\alpha}}^n \in C_\delta$ such that

$$\left\| \sum_{l=1}^m \alpha_l^n \phi_H^l - \sum_{l=1}^m \tilde{\alpha}_l^n \widetilde{\phi}_H^l \right\|_\infty$$
$$\leq \sum_{l=1}^m |\alpha_l^n| \left\| \phi_H^l - \widetilde{\phi}_H^l \right\|_\infty + \sum_{l=1}^m |\widetilde{\alpha}_l^n - \alpha_l^n| \left\| \widetilde{\phi}_H^l \right\|_\infty$$

$$\leq \sum_{l=1}^m \alpha_l^n \delta + \sum_{l=1}^m |\widetilde{\alpha}_l^n - \alpha_l^n| \leq 2\delta,$$

and for any $(o_h, a_h) \in \mathcal{O} \times \mathcal{A}$

$$\left\| \sum_{l=1}^m \alpha_l^n \mathbf{M}_h^l - \sum_{l=1}^m \widetilde{\alpha}_l^n \widetilde{\mathbf{M}}_h^l \right\|_\infty$$

$$\leq \sum_{l=1}^m |\alpha_l^n| \left\| \mathbf{M}_h^l - \widetilde{\mathbf{M}}_h^l \right\|_\infty + \sum_{l=1}^m |\widetilde{\alpha}_l^n - \alpha_l^n| \left\| \widetilde{\mathbf{M}}_h^l \right\|_\infty$$

$$\leq \sum_{l=1}^m \alpha_l^n \sqrt{r} \left\| \mathbf{Vec}(\mathbf{M}_h^l) - \mathbf{Vec}(\widetilde{\mathbf{M}}_h^l) \right\|_\infty + \sum_{l=1}^m |\widetilde{\alpha}_l^n - \alpha_l^n| \left\| \widetilde{\mathbf{M}}_h^l \right\|_\infty$$

$$\leq \sum_{l=1}^m \alpha_l^n \sqrt{r}\delta + \sum_{l=1}^m \sqrt{r}|\tilde{\alpha}_l^n - \alpha_l^n|$$

$$= \sqrt{r}\delta + \sqrt{r} \left\| \tilde{\alpha}^n - \alpha^n \right\|_1 \leq 2\sqrt{r}\delta.$$

Similar to the analysis in the proof of Lemma 6, specifically, Equation (15), $\widetilde{\Theta}_\delta^t$ can constitute an $\eta$-bracket for $\Theta^t$ with $\delta = \frac{\eta}{2\sqrt{r}(|\mathcal{O}||\mathcal{A}|)^{cH}}$.

The cardinality of the cover of multi-task model class is $|\widetilde{\Theta}_\delta| = |\widetilde{\Theta}_\delta^0||C_\delta|^N = (3\sqrt{r} \times \frac{1}{\delta})^{r^2 H^2 |\mathcal{O}||\mathcal{A}|m} \times (\frac{3}{\delta})^{mN}$.

In conclusion, the log $\eta$-bracketing number is $O\left( r^2 H^2 |\mathcal{O}||\mathcal{A}|m \log(\frac{rH|\mathcal{O}||\mathcal{A}|}{\eta}) + mN \log(\frac{rH|\mathcal{O}||\mathcal{A}|}{\eta}) \right)$.

$\square$

### E.3 BRACKETING NUMBER OF DOWNSTREAM EXAMPLES

**Example 1** Note that the model class for downstream learning is $\hat{\Theta}_0^u = \{\mathbb{O}_h\}_{h \in [H]}$. Thus, we immediately obtaint that $\log \mathcal{N}_\eta(\hat{\Theta}_0^u) = O(H|\mathcal{O}||\mathcal{S}| \log \frac{|\mathcal{O}||\mathcal{S}|}{\eta})$

**Example 2(Multi-task PSR with perturbed models):** Similar to the upstream tasks, the downstream task 0 is also a noisy perturbation of the latent base task. Specifically, for each step $h \in [H]$, any $(o, a) \in \mathcal{O} \times \mathcal{A}$, we have

$$\phi_H^0 = \phi_H^b, \mathbf{M}_h^0(o_h, a_h) = \mathbf{M}_h^b(o_h, a_h) + \Delta_h^0(o_h, a_h), \quad \Delta_h^0 \in \mathbf{\Delta}. \tag{16}$$

The log $\eta$-bracketing number is at most $H \log |\mathbf{\Delta}|$.

*Proof.* Suppose the estimated model parameter of the base task is $\bar{\theta}^b = \{\bar{\phi}_H^b, \{\overline{\mathbf{M}}_h^b\}_{h=1}^H\}$. Because the downstream task model parameters satisfy Equation (16), the empirical candidate model class can be characterized as

$$\hat{\Theta}_0^u = \{\theta : \theta = \{\phi_H^0, \{\mathbf{M}_h^0\}_{h=1}^H\}; \phi_H^0 = \bar{\phi}_H^b;$$
$$\mathbf{M}_h^0(o_h, a_h) = \overline{\mathbf{M}}_h^b(o_h, a_h) + \Delta_h^0(o_h, a_h), (o_h, a_h) \in \mathcal{O} \times \mathcal{A}; \Delta_h^0 \in \mathbf{\Delta} \}.$$

If $\bar{\theta}^b$ is given, then the candidate model class is decided by $\mathbf{\Delta}$. Then for any $\eta > 0$, the $\eta$-bracketing number is $|\mathbf{\Delta}|^H$. $\square$

**Example 3(Multi-task PSRs: Linear combination of core tasks):** Suppose the downstream task 0 also lies in the linear span of $m$ core tasks same as the upstream. Moreover, assume the upstream tasks are diverse enough to span the whole core tasks space. As a result, the downstream task can be represented as a linear combination of a subset of the upstream tasks. Specifically, there exists a constant $L$ satisfying $m \leq L \leq N$ and a coefficient vector $\boldsymbol{\alpha}^0 = (\alpha_1^0, \cdots, \alpha_L^0)^\top \in \mathbb{R}^L$ s.t. for any $h \in [H]$ and $(o_h, a_h) \in \mathcal{O} \times \mathcal{A}$,

$$\phi_H^0 = \sum_{l=1}^L \alpha_l^0 \phi_H^l, \quad \mathbf{M}_h^0(o_h, a_h) = \sum_{l=1}^L \alpha_l^0 \mathbf{M}_h^l(o_h, a_h). \tag{17}$$

For regularization, we assume $0 \le \alpha_l^0 \le 1$ for all $l \in [L]$, and $\sum_{l=1}^{L} \alpha_l^0 = 1$. It can be shown that $\beta_0 = O(LH \log(\frac{r|\mathcal{O}||\mathcal{A}|}{\eta}))$, whereas $\beta^{(1)}$ for learning without prior information is given by $O(r^2|\mathcal{O}||\mathcal{A}|H^2 \log \frac{|\mathcal{O}||\mathcal{A}|}{\eta})$. Clearly, $\beta_0 \ll \beta^{(1)}$ holds if $m \le \min\{N, r^2|\mathcal{O}||\mathcal{A}|H^2\}$, which is satisfied in practice.

*Proof.* Suppose for the upstream task $l \in [L]$, the estimated model parameter is $\bar{\theta}_l = \{\bar{\phi}_H^l, \{\overline{\mathbf{M}}_h^l\}_{h=1}^H\}$. Because the downstream task model parameters satisfy Equation (17), the empirical candidate model class can be characterized as

$$\hat{\Theta}_0^u = \{\theta : \theta = \{\phi_H^0, \{\mathbf{M}_h^0\}_{h=1}^H\}; \phi_H^0 = \sum_{l=1}^{L} \alpha_n^0 \bar{\phi}_H^l;$$

$$\mathbf{M}_h^0(o_h, a_h) = \sum_{l=1}^{L} \alpha_l^0 \overline{\mathbf{M}}_h^l(o_h, a_h), (o_h, a_h) \in \mathcal{O} \times \mathcal{A}; \boldsymbol{\alpha}^0 \in \mathbb{R}^L \}.$$

Then we consider the cover for $\hat{\Theta}_0^u$. Denote $\boldsymbol{C}_\delta^L$ as a $\delta$-cover for the unit ball in $\mathbb{R}^L$ w.r.t. $\ell_1$ norm. Mathematically, for any vector $\boldsymbol{\alpha}^0 \in \mathbb{R}^L$, there exists an $\tilde{\boldsymbol{\alpha}}^0 \in \boldsymbol{C}_\delta^L$ such that $\left\| \boldsymbol{\alpha}^0 - \tilde{\boldsymbol{\alpha}}^0 \right\|_1 \le \delta$. In addition, the cardinality $|\boldsymbol{C}_\delta^L| = (\frac{3}{\delta})^L$.

Define

$$\widetilde{\Theta}_\delta^u = \{\theta : \theta = \{\phi_H^0, \{\mathbf{M}_h\}_{h=1}^H\}; \phi_H^0 = \sum_{l=1}^{L} \tilde{\alpha}_l^0 \bar{\phi}_H^l;$$

$$\mathbf{M}_h^0(o_h, a_h) = \sum_{l=1}^{L} \tilde{\alpha}_l^0 \overline{\mathbf{M}}_h^l(o_h, a_h), (o_h, a_h) \in \mathcal{O} \times \mathcal{A}; \tilde{\boldsymbol{\alpha}}^0 \in \boldsymbol{C}_\delta^L \}.$$

Then, for any $\theta^0 \in \hat{\Theta}_0$ with $\boldsymbol{\alpha}^0 \in \mathbb{R}^L$, there exist $\tilde{\theta} \in \tilde{\Theta}_\delta^u$ with $\tilde{\boldsymbol{\alpha}}^0 \in \boldsymbol{C}_\delta^L$ such that

$$\left\| \sum_{l=1}^{L} \alpha_l^0 \bar{\phi}_H^l - \sum_{l=1}^{L} \tilde{\alpha}_l^0 \bar{\phi}_H^l \right\|_\infty \le \sum_{l=1}^{L} |\alpha_l^0 - \tilde{\alpha}_l^0| \left\| \bar{\phi}_H^l \right\|_\infty \le \left\| \boldsymbol{\alpha}^0 - \tilde{\boldsymbol{\alpha}}^0 \right\|_1 \le \delta,$$

and for any $(o_h, a_h) \in \mathcal{O} \times \mathcal{A}$

$$\left\| \sum_{l=1}^{L} \alpha_l^0 \overline{\mathbf{M}}_h^l(o_h, a_h) - \sum_{l=1}^{L} \tilde{\alpha}_l^0 \overline{\mathbf{M}}_h^l(o_h, a_h) \right\|_\infty$$

$$\le \sum_{l=1}^{L} |\alpha_l^0 - \tilde{\alpha}_l^0| \left\| \overline{\mathbf{M}}_h^l(o_h, a_h) \right\|_\infty \le \sqrt{r} \left\| \boldsymbol{\alpha}^0 - \tilde{\boldsymbol{\alpha}}^0 \right\| \le \sqrt{r}\delta.$$

Similar to the analysis in the proof of Lemma 6, specifically, Equation (15), $\widetilde{\Theta}_\delta^u$ can constitute an $\eta$-bracket for $\hat{\Theta}_0^u$ with $\delta = \frac{\eta}{2\sqrt{r}(|\mathcal{O}||\mathcal{A}|)^{cH}}$.

The cardinality of the cover of multi-task model class is $|\widetilde{\Theta}_\delta^u| = |\boldsymbol{C}_\delta^L| = (\frac{3}{\delta})^L$. In conclusion, the log $\eta$-bracketing number is $O\left(LH \log(\frac{r|\mathcal{O}||\mathcal{A}|}{\eta})\right)$. □

# F  EXAMPLES OF MULTI-TASK MDPS FROM PREVIOUS WORK

To demonstrate that our framework encompasses multi-task learning under MDPs, we provide several examples of MDPs from previous work in this subsection. Suppose $\mathcal{S}$ is the state space, there exist $N$ source tasks, and $P^{(*,n)} : \mathcal{S} \times \mathcal{A} \times \mathcal{S} \to \mathbb{R}$ is the true transition kernel of task $n$.

Cheng et al. (2022) studied multi-task learning under low-rank MDPs in which the transition kernel $P^{(*,n)}$ has a $d$ dimension low-rank decomposition into two embedding functions $\phi^{(*)} : \mathcal{S} \times \mathcal{A} \to$

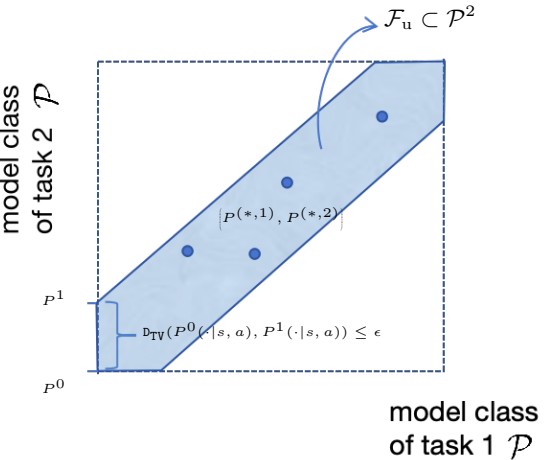

Figure 2: Supplementary illustration of joint model class in two dimensions for MDPs. A concrete example of tasks with similar transition kernels, i.e., any point $(P^0, P^1)$ in the joint class $\mathcal{F}_u$ satisfies $\max_{(s,a)\in\mathcal{S}\times\mathcal{A}} D_{TV}(P^0(\cdot|s,a), P^1(\cdot|s,a)) \leq \epsilon$ for a small positive constant $\epsilon$.

$\mathbb{R}^d, \mu^{(*,n)} : \mathcal{S} \to \mathbb{R}^d$ as $P^{(*,n)}(s,a,s') = \langle \phi^{(*)}(s,a), \mu^{(*,n)}(s') \rangle$ for all $(s,a,s') \in \mathcal{S}\times\mathcal{A}\times\mathcal{S}$ for each task $n$. In this setting, Cheng et al. (2022) assume the $N$ source tasks share common representations $\phi^{(*)}$ and for each task $n$, $\phi^{(*)} \in \Phi, \mu^{(*,n)} \in \Psi$ for finite model class $\{\Phi, \Psi\}$. Consequently, the $\eta$-bracketing number for the multi-task low-rank MDPs model class is at most $O(H|\Phi||\Psi|^N)$, which is much smaller than the one of the individual single-task with $O(H|\Phi|^N|\Psi|^N)$ if $|\Phi| \gg |\Psi|$.

Zhang & Wang (2021) studied multi-task learning under tabular MDPs with an assumption that for any two tasks $n_1, n_2 \in [N]$, it holds that $\max_{(s,a)\in\mathcal{S}\times\mathcal{A}} D_{TV}(P^{(*,n_1)}(\cdot|s,a)|P^{(*,n_2)}(\cdot|s,a)) \leq \epsilon$ for some small $\epsilon > 0$. Then the multi-task model class is much smaller than the model class of $N$ individual single-task (see Figure 2 for an illustration when $N = 2$). Consequently, the $\eta$-bracketing number for the multi-task low-rank MDPs model class is at most $O(H|\mathcal{S}|^2|\mathcal{A}|(\log \frac{H|\mathcal{A}||\mathcal{S}|}{\eta} + (N-1)\log \frac{H|\mathcal{A}||\mathcal{S}|\epsilon}{\eta}))$. This is smaller than that of each individual single-task, which is $O(HN|\mathcal{S}|^2|\mathcal{A}|\log \frac{H|\mathcal{A}||\mathcal{S}|}{\eta})$ if $\epsilon \leq \eta$ and $N \geq 1$.

## G AUXILLARY LEMMAS

The following lemma characterizes the relationship between the total variation distance and the Hellinger-squared distance. Note that the result for probability measures has been proved in Lemma H.1 in Zhong et al. (2022). Since we consider more general bounded measures, we provide the full proof for completeness.

**Lemma 7.** *Given two bounded measures $P$ and $Q$ defined on the set $\mathcal{X}$, let $|P| = \sum_{x\in\mathcal{X}} P(x)$ and $|Q| = \sum_{x\in\mathcal{X}} Q(x)$. We have*

$$D_{TV}^2(P,Q) \leq 4(|P| + |Q|)D_H^2(P,Q).$$

*In addition, if $P_{Y|X}, Q_{Y|X}$ are two conditional distributions over a random variable $Y$, and $P_{X,Y} = P_{Y|X}P, Q_{X,Y} = Q_{Y|X}Q$ are the joint distributions when $X$ follows the distributions $P$ and $Q$, respectively, we have*

$$\mathbb{E}_{X\sim P}\left[D_H^2(P_{Y|X}(\cdot|X), Q_{Y|X}(\cdot|X))\right] \leq 8D_H^2(P_{X,Y}, Q_{X,Y}).$$

**Lemma 8.** *Suppose $\mathbb{P}$ and $\mathbb{Q}$ are two probability distributions. For any $\alpha > 1$, we have the following inequality.*

$$D_{TV}(\mathbb{P},\mathbb{Q}) \leq \sqrt{\frac{1}{2}D_{R,\alpha}(\mathbb{P},\mathbb{Q})}.$$

*Proof.* By Pinsker's inequality, we have

$$\mathrm{D}_{\mathrm{TV}}(\mathbb{P}, \mathbb{Q}) \leq \sqrt{\frac{1}{2} \mathrm{D}_{\mathrm{KL}}(\mathbb{P}, \mathbb{Q})}.$$

By Theorem 5 from Van Erven & Harremos (2014), we have

$$\mathrm{D}_{\mathrm{KL}}(\mathbb{P}, \mathbb{Q}) = \lim_{\alpha \uparrow 1} \mathrm{D}_{\mathrm{R}, \alpha}(\mathbb{P}, \mathbb{Q}) \leq \inf_{\alpha > 1} \mathrm{D}_{\mathrm{R}, \alpha}(\mathbb{P}, \mathbb{Q}).$$

$\square$

**Lemma 9** (Elliptical potential lemma ). *For any sequence of vectors $\mathcal{X} = \{x_1, \ldots, x_n, \ldots\} \subset \mathbb{R}^d$, let $U_k = \lambda I + \sum_{t < k} x_k x_k^\top$, where $\lambda$ is a positive constant, and $B > 0$ is a real number. If the rank of $\mathcal{X}$ is at most $r$, then, we have*

$$\sum_{k=1}^K \min\left\{ \|x_k\|_{U_k^{-1}}^2, B \right\} \leq (1 + B) r \log(1 + K/\lambda),$$

$$\sum_{k=1}^K \min\left\{ \|x_k\|_{U_k^{-1}}, \sqrt{B} \right\} \leq \sqrt{(1 + B) r K \log(1 + K/\lambda)}.$$

*Proof.* Note that the second inequality is an immediate result from the first inequality by the Cauchy's inequality. Hence, it suffices to prove the first inequality. To this end, we have

$$\sum_{k=1}^K \min\left\{ \|x_k\|_{U_k^{-1}}^2, B \right\} \overset{(i)}{\leq} (1 + B) \sum_{k=1}^K \log\left(1 + \|x_k\|_{U_k^{-1}}^2\right)$$

$$= (1 + B) \sum_{k=1}^K \log\left(1 + \texttt{trace}\left((U_{k+1} - U_k) U_k^{-1}\right)\right)$$

$$= (1 + B) \sum_{k=1}^K \log\left(1 + \texttt{trace}\left(U_k^{-1/2} (U_{k+1} - U_k) U_k^{-1/2}\right)\right)$$

$$\leq (1 + B) \sum_{k=1}^K \log \texttt{det}\left(I_d + U_k^{-1/2} (U_{k+1} - U_k) U_k^{-1/2}\right)$$

$$= (1 + B) \sum_{k=1}^K \log \frac{\texttt{det}(U_{k+1})}{\texttt{det}(U_k)}$$

$$= (1 + B) \log \frac{\texttt{det}(U_{K+1})}{\texttt{det}(U_1)}$$

$$= (1 + B) \log \texttt{det}\left(I + \frac{1}{\lambda} \sum_{k=1}^K x_k x_k^\top\right)$$

$$\overset{(ii)}{\leq} (1 + B) r \log(1 + K/\lambda),$$

where $(i)$ follows because $x \leq (1 + B) \log(1 + x)$ if $0 < x \leq B$, and $(ii)$ follows because $\texttt{rank}(\mathcal{X}) \leq r$. $\square$

To compute the bracketing number of mulit-task model class, we first require a basic result on the covering number of a Euclidean ball as follows. Proof of the lemma can be found in Lemma 5.2 in Vershynin (2010).

**Lemma 10** (Covering Number of Euclidean Ball). *For any $\epsilon > 0$, the $\epsilon$-covering number of the Euclidean ball in $\mathbb{R}^d$ with radius $R > 0$ is upper bounded by $(1 + 2R/\epsilon)^d$.*

