# OpenReview forum: "Provable Benefits of Multi-task RL under Non-Markovian Decision Making Processes"
_ICLR.cc/2024/Conference — ICLR 2024 poster_

### Official Review · Reviewer_FV3A · 2023-10-27

**Soundness:** 2 fair
**Presentation:** 2 fair
**Contribution:** 2 fair
**Rating:** 6
**Confidence:** 1

**Summary:**

This paper studied multi-task reinforcement learning (MTRL) in complex environments like partially observable MDPs and predictive state representations. The authors identified two main challenges: 1. Identifying Beneficial Common Latent Structures: The large and complex model space of multi-task predictive state representations (PSRs) made it difficult to identify types of common latent structures that could reduce model complexity. 2. Intertwining of Model Learning and Data Collection: in RL, model learning and data collection are intertwined, creating temporal dependencies in the collected data. This complicates the analysis of multi-task PSRs, making it challenging to gauge the benefits of reduced model complexity in terms of statistical efficiency gains in RL. To solve these challenges, the authors introduced the η-bracketing number to quantify model complexity and task similarity. They developed the UMT-PSR algorithm for efficient upstream multi-task learning and addressed downstream transfer learning by leveraging similarities with previously learned tasks. Their contributions include a new complexity metric, the innovative UMT-PSR algorithm, and techniques for enhanced downstream learning, marking a pioneering theoretical exploration of multi-task RL's benefits over single-task approaches in complex environments.

**Strengths:**

To AC: I do not have the expertise in this research area to review the strengths and weaknesses of the paper. Please lower the weight of my review.

**Weaknesses:**

To AC: I do not have the expertise in this research area to review the strengths and weaknesses of the paper. Please lower the weight of my review.

**Questions:**

To AC: I do not have the expertise in this research area to review the strengths and weaknesses of the paper. Please lower the weight of my review.

---

> ### Author Response · Authors · 2023-11-19
> **Response to Reviewer FV3A**
>
> We thank the reviewer for reading and summarizing the paper. We also thank the reviewer for acknowledging the reviewer’s lack of expertise in this area. Given that the reviewer does not express any negative concerns about the paper and the general opinion from the other reviewers is unanimously positive, we humbly ask the reviewer to kindly consider to raise the rating. Thanks again!

---

> > ### Comment · Reviewer_FV3A · 2023-11-22
> > **score update**
> >
> > I am not knowledgeable in this area of research. I have adjusted my score based on the general positive opinion of all other reviews.

---

> > > ### Author Response · Authors · 2023-11-22
> > >
> > > We thank the reviewer very much for kindly increasing the score.

---

### Official Review · Reviewer_qpJY · 2023-10-31

**Soundness:** 3 good
**Presentation:** 4 excellent
**Contribution:** 2 fair
**Rating:** 6
**Confidence:** 4

**Summary:**

This paper studies the benefits of multi-task learning in low-rank PSRs when tasks share similar latent structures. It proposes a measurement for the similarity of $N$ tasks called the $\eta$-bracketing number, which is shown to be small in several standard classes. Examples include multi-task POMDPs sharing the state space, action space, and transition kernel, and multi-task low-rank PSRs with similar core set observable matrices. The algorithm proposed in the paper leverages the $\eta$-bracketing number to find the optimistic exploration policies, and follows the idea of MLE to build confidence set for models. It is proved to gain benefits when the average $\eta$-bracketing number over $N$ tasks is smaller than the $\eta$-bracketing number for a single task in terms of sample complexity to find the optimal policies of all tasks. Additionally, the paper also gives a downstream algorithm that improves the sample complexity to identify a new target task based on the similarity between target task and the $N$ original tasks.

**Strengths:**

1. The setting is an extension of multi-task learning of MDPs and bandits, which is novel in the literature of multi-task decision making.

2. It provides a key measurement of similarity in the multi-task setting, the $\eta$-bracketing number, to identify the effectiveness of the multi-task transfer learning in low-rank PSRs.

**Weaknesses:**

1. The results of the paper is interesting in the viewpoint of setting and techniques, but not surprising given many previous works on multi-task reinforcement learning. The standard tool to establish the benefits of shared structure of multiple tasks is the reduced covering number of the joint model class or value class. For example, the common low-rank assumption in the linear setting essentially reduce the log covering number of the function class from $nm$ to $nk + km$, where $n, m, k$ denotes the ambient dimension, the number of tasks, and the rank that is small. This work studies a more complicated setting, but the $\eta$-bracketing number is essentially some type of covering number over the joint model class (see the questions below). As long as this key property is identified, the remaining task is to follow OMLE to perform optimistic planning in the joint model space.

2. The generality of the theorems in the paper allows the various common structure with different $\eta$-bracketing number of the joint model class. Several examples are already explained in the paper such as the multi-task observable POMDP sharing the common transition kernel. The generic upstream algorithm is highly computational inefficient in building the confidence set and find the optimistic policy. Therefore, an important question is how the optimization steps in the algorithm look like in a specific setting (e.g., the multi-task POMDPs). This helps to evaluate the effectiveness of the upstream algorithms.

**Questions:**

It seems that the $\eta$-bracketing number is used to build the optimism of optimistic policies with uniform convergence over the bracketing set. This is essentially (and also used as) the covering number of the model class in terms of $\|\cdot\|^{\mathrm{p}}_{\infty}$ norm. Why use bracketing number as the name instead of calling it the covering number directly?

---

> ### Author Response · Authors · 2023-11-19
> **Response to Reviewer qpJY (Part 1 /2)**
>
> We thank the reviewer for the careful reading and thoughtful comments. We address the reviewer's questions in the following and revised the paper accordingly. The changes are marked in blue   in our revision. We hope the responses below and changes in the paper address the reviewers' concerns.
>
>
> **Q1:** The results of the paper is interesting in the viewpoint of setting and techniques, but not surprising given many previous works on multi-task reinforcement learning. The standard tool to establish the benefits of shared structure of multiple tasks is the reduced covering number of the joint model class or value class. For example, the common low-rank assumption in the linear setting essentially reduce the log covering number of the function class from $nm$ to $nk+km$, where $n,m,k$ denotes the ambient dimension, the number of tasks, and the rank that is small. This work studies a more complicated setting, but the $\eta$-bracketing number is essentially some type of covering number over the joint model class (see the questions below). As long as this key property is identified, the remaining task is to follow OMLE to perform optimistic planning in the joint model space.
>
> A1: We would like to highlight our new analytical development. (1) Even after the identification of the bracketing number as an appropriate measure of complexity, using it to develop performance guarantees for multi-task OMLE is not straightforward, and requires new technical developments. For example, our optimistic planning is different from OMLE since Algorithm 1 chooses a pairwise additive distance uniquely designed for the multi-task case. Then upper bounding the total variation distance based on such a pairwise additive distance requires new analysis beyond that used in OMLE. (2) Our study in downstream learning also requires new analysis. Since the downstream model class is not known in advance, the agent needs to use information from the upstream to approximate the model, which introduces an approximation error $\epsilon_0$. Our handling of such an error here features a novel technique of using the properties of the **R\'enyi divergence** to quantify the approximation error without requiring the realizability assumption, which guarantees the sub-optimality bound **without requiring the realizability condition**. Such a technique has not been developed in previous results.
>
> **Q2**: The generality of the theorems in the paper allows the various common structure with different $\eta$-bracketing number of the joint model class. Several examples are already explained in the paper such as the multi-task observable POMDP sharing the common transition kernel. The generic upstream algorithm is highly computational inefficient in building the confidence set and find the optimistic policy. Therefore, an important question is how the optimization steps in the algorithm look like in a specific setting (e.g., the multi-task POMDPs). This helps to evaluate the effectiveness of the upstream algorithms.
>
> **A2:** The optimization step in our algorithm (in Line 3) contains two components: (1) The optimization over the model parameter  $\boldsymbol{\theta},\boldsymbol{\theta}'$ subject to the constraints that define the confidence set $\boldsymbol{\mathcal{B}}_k$ in Line 7 of Algorithm 1. Note that the maximization in the constraints in Line 7 will need one inner-loop of optimization to solve to define the constraint thresholds. For a given specific setting, certain function approximations (e.g., neural networks) can be used to parameterize the model transitions in an efficient way. Then the standard constrained optimization algorithms such as primal-dual method can be applied to solve the problem. (2) The optimization over policy can be done by first parameterizing policy and then using policy gradient methods. Alternatively, some papers in practice use tree search and sample-based methods to solve policy optimization in POMDPs [5, 6, 7].

---

> > ### Author Response · Authors · 2023-11-19
> > **Response to Reviewer qpJY (Part 2 /2)**
> >
> > **Q3:** It seems that the $\eta$-bracketing number is used to build the optimism of optimistic policies with uniform convergence over the bracketing set. This is essentially (and also used as) the covering number of the model class in terms of $|\cdot|_{\infty}^p$ norm. Why use bracketing number as the name instead of calling it the covering number directly?
> >
> > **A3:** Great question! Bracketing number is more suitable in our study for providing two-sided bounds (upper and lower bounds) to control the total variation distance between model classes, but we don't see a direct way to use the covering number for serving this purpose. Specifically, the bracketing number is used for univariate- or vector-valued functions classes, which satisfies certain partial order relations. Based on the partial order relation, the bracketing set requires a "two-sided" and "bracketing type" covering to control both upper  and lower bounds of the elements from target class, which is useful in analysis to control the total variation distance. In contrast, the covering set does not provide  upper and lower bounds and only require being close to elements from target class. Mathematically, for a vector-valued functions $\mathbf{f}: \mathcal{X} \to \mathbb{R}^d$, the $\epsilon$-bracketing number requires a bracket $[\mathbf{l},\mathbf{g}]$ such that $\mathbf{l}\leq\mathbf{f}\leq\mathbf{g}$ and $\||\mathbf{l}-\mathbf{g}\||\leq \epsilon$. However, the $\epsilon$-covering number only requires a ball with center $\mathbf{r}$ such that $\||\mathbf{r}-\mathbf{f}\||\leq \epsilon$.
> >
> > [1] Zhan, Wenhao, et al. "PAC Reinforcement Learning for Predictive State Representations." The Eleventh International Conference on Learning Representations. 2022.
> >
> > [2] Qinghua Liu and Praneeth Netrapalli and Csaba Szepesvári and Chi Jin. Optimistic MLE -- A Generic Model-based Algorithm for Partially Observable Sequential Decision Making. arXiv preprint
> > arXiv:2209.14997, 2022.
> >
> > [3] Kaiser et al. Model Based Reinforcement Learning for Atari. ICLR 2020
> >
> > [4] Chen, Fan, Yu Bai, and Song Mei. "Partially Observable RL with B-Stability: Unified Structural Condition and Sharp Sample-Efficient Algorithms." The Eleventh International Conference on Learning Representations. 2022.
> >
> > [5] Wu, C., Yang, G., Zhang, Z., Yu, Y., Li, D., Liu, W., Hao, J. (2021). Adaptive online packing-guided search for POMDPs. Advances in Neural Information Processing Systems, 34, 28419-28430.
> >
> > [6] Sunberg, Z., Kochenderfer, M. (2018, June). Online algorithms for POMDPs with continuous state, action, and observation spaces. In Proceedings of the International Conference on Automated Planning and Scheduling (Vol. 28, pp. 259-263).
> >
> > [7] Hoerger, M., Kurniawati, H. (2021, May). An on-line POMDP solver for continuous observation spaces. In 2021 IEEE International Conference on Robotics and Automation (ICRA) (pp. 7643-7649). IEEE.

---

### Official Review · Reviewer_W5uL · 2023-10-31

**Soundness:** 3 good
**Presentation:** 2 fair
**Contribution:** 3 good
**Rating:** 8
**Confidence:** 3

**Summary:**

This theoretical paper discusses the problem of multi-task reinforcement learning in the setting of low-rank and well-conditioned PSRs. It analyzes two situations when there is structure in the multiple tasks: upstream learning and downstream.

**Strengths:**

Strengths of this paper are that it tackles a challenging problem, introduces a nice formulation for the problem and showcases interesting theoretical results on when multi-task learning is beneficial as opposed to multiple single-task learning.

**Weaknesses:**

The biggest weakness of this paper is that it crams too much content in the main paper, and does not use the appendix to explicate it. This rushed discussion makes the job of reading the paper more difficult than it needs to be. For example, consider the assumptions of rank-$r$ and $\gamma$-well-conditioned PSRs. These two assumptions are present throughout the paper, but they never get their deserved attention. A mere half a page of terse definitions for them might be justified in the main paper due to page number constraints, but it is hard to justify why they were never given due discussion in the appendix. This discussion should discuss the intuitive meaning of these assumptions, examples of PSRs which satisfy the assumptions (otherwise questions like, is the set $\Theta$ even non-empty, surface), examples of PSRs which don't satisfy the assumptions, what fails in the proofs if each assumption is relaxed, etc. As for other examples, the notation $\phi_h$ and $\mathbf{M} _ h$ is never even defined, the norm $\lVert \cdot \rVert_\infty^p$  is "pulled out of a hat", etc.

The paper does a poor job at literature review. For example, it states that "none of the existing studies considered multi-task POMDPs/PSRs", but consider _Multi-task Reinforcement Learning in Partially Observable Stochastic Environment_ by Li, Liao and Carin (JMLR 2009) or _Deep Decentralized Multi-task Multi-Agent Reinforcement Learning under Partial Observability_ by Omidshafiei, Pazis, Amato, How and Vian (ICML 2017).

While not necessarily a weakness, it would have been nice to demonstrate the usefulness of the theory developed in the paper on some simple experiments. As long as I am asking for things it would be nice to calculate the computational complexity of implementing the algorithms. Note that I am not at all expecting these additions to this paper -- the paper is already very terse as it is.

**Questions:**

1. Why is the first goal of upstream learning finding near-optimal policies for all $N$ tasks _on average_, as opposed to, say, finding near-optimal policies for all $N$ tasks, i.e.
$$
\begin{align*}
max_{n \in [N]} \max_\pi \left(V_{\theta_n^*,\,R_n}^{\pi} - V_{\theta_n^*,\,R_n}^{\bar{\pi}^n}\right) \le \epsilon.
\end{align*}
$$
2. Why use $\lVert \cdot \rVert_\infty^p$ as the norm? Is it even a norm (i.e., satisfies the conditions required)?
3. In the calculation of $\eta$-bracketing number of $\{(\mathbb{P} _ {\theta_1}, \ldots, \mathbb{P} _ {\theta_N}) : \mathbf{\theta} \in \mathbf{\Theta} _ u\}$, what is the domain of the functions? Is it the $\sigma-$algebra over $(\mathcal{O} \times \mathcal{A})^H,$ which is the domain of the distributions? Consider a simpler calculation: how to calculate the $\eta$-bracketing number for $\{\mathbb{P} _ {\theta} : \theta \in \Theta\}$? Now $\mathbb{P} _ {\theta}$ is a probability measure which is defined over some $\sigma$-algebra $\mathscr{S}$. If it is contained in some $\eta$-bracket $[\mathbb{A}, \mathbb{B}]$, then we must have $\mathbb{A}(S) \leq \mathbb{P} _ \theta(S) \le \mathbb{B}(S)$ for every $S \in \mathscr{S}$. But this would imply (it might require measures to be regular, I am not sure) that $\mathbb{A} = \mathbb{P} _ \theta = \mathbb{B}$. So the $\eta$-bracketing number for $\{\mathbb{P} _ {\theta} : \theta \in \Theta\}$ becomes $|\Theta|$. I am assuming that this is not what the authors had in mind. Could you please clarify the calculation? The calculations in Appendix E are assuming the observation and action spaces are finite.
4. I do not understand the discussion of pairwise additive distance based multi-task planning. Why is a distance between product distributions not sufficient as opposed to what the paper uses? Also, do the authors realize that
   $$
  \sum _ {n \in [N]}\mathtt{D _ {TV}}(\mathbb{P} _ {\theta_n}, \mathbb{P} _ {\theta'_n}) = \mathtt{D _ {TV}}(\mathbb{P} _ {\theta_1} \otimes \cdots \otimes \mathbb{P} _ {\theta_N}, \mathbb{P} _ {\theta'_1} \otimes \cdots \otimes \mathbb{P} _ {\theta'_N}),
  $$
   a divergence (not distance) over product distribution?

---

> ### Author Response · Authors · 2023-11-19
> **Response to Reviewer W5uL (Part 1/2)**
>
> We thank the reviewer for the careful reading and thoughtful comments. We address the reviewer's questions in the following and revised the paper accordingly. The changes are marked in blue in our revision. We hope the responses below and changes in the paper address the reviewers' concerns.
>
> **Q1:** The biggest weakness of this paper is that it crams too much content in the main paper, and does not use the appendix to explicate it.  For example, consider the assumptions of rank-$r$
>  and
> $\gamma$-well-conditioned PSRs. These two assumptions are present throughout the paper, but they never get their deserved attention. A mere half a page of terse definitions for them might be justified in the main paper due to page number constraints, but it is hard to justify why they were never given due discussion in the appendix. This discussion should discuss the intuitive meaning of these assumptions, examples of PSRs which satisfy the assumptions (otherwise questions like, is the set
>  even non-empty, surface), examples of PSRs which don't satisfy the assumptions, what fails in the proofs if each assumption is relaxed, etc. As for other examples, the notation $\phi\_h$
>  and $\mathbf{M}\_h$
>  is never even defined, the norm $\\|\cdot\\|\_{\infty}^{\mathrm{p}}$
>  is "pulled out of a hat", etc.
>
> **A1:** Thanks for the careful reading. We humbly acknowledge that the content in the main text is rather dense. If this has hinder the reviewer's understanding of our work, we sincerely apologize. If this paper is accepted, we will streamline the paper to make sure it is more readable and provide intuitions for, e.g., the $\gamma$-well-conditioned PSRs, in the Appendix following the reviewer's suggestions.
>
> Briefly speaking, the   **$\gamma$-well condition** property of PSRs ensures that error amplification which results from estimating $\psi^*(\tau\_h)$ is not extremely large since otherwise,  there exists a hard instance that is not well-conditioned and learning a good policy requires an exponentially large sample complexity [1]. The definition of $\phi\_h$
>  and $\mathbf{M}\_h$ can be found on page 4, between Eqns. (1) and (2). The definition of the norm $\\|\cdot\\|\_{\infty}^{\mathrm{p}}$ (which depends on policy $\mathrm{p}$) can be found in the 4th paragraph on page 5. We have ensured that all mathematical symbols are defined before use, although due to the density of results, they may not be very prominently displayed.
>
> **Q2:** The paper does a poor job at literature review. For example, it states that "none of the existing studies considered multi-task POMDPs/PSRs", but consider Multi-task Reinforcement Learning in Partially Observable Stochastic Environment by Li, Liao and Carin (JMLR 2009) or Deep Decentralized Multi-task Multi-Agent Reinforcement Learning under Partial Observability by Omidshafiei, Pazis, Amato, How and Vian (ICML 2017).
>
> **A2:** Thanks for the kind suggestions. By "none of the existing studies considered multi-task POMDPs/PSRs", we meant "with theoretical performance characterization". We will clarify this in our paper. Meanwhile, we agree with the reviewer that we should do a more extensive literature review. In the final version of the paper, we will do a more  detailed literature review on this topic. In the meantime, we have added these references and revised the paper accordingly. The reviewer is welcome to check the revised version of the manuscript.
>
>
> **Q3:** While not necessarily a weakness, it would have been nice to demonstrate the usefulness of the theory developed in the paper on some simple experiments. As long as I am asking for things it would be nice to calculate the computational complexity of implementing the algorithms. Note that I am not at all expecting these additions to this paper -- the paper is already very terse as it is.
>
> **A3:** Thanks for the kind suggestions. This paper is primarily positioned as a theoretical contribution. If we can find a suitable dataset, we will conduct  some experiments and report the results in the final version of the paper.
>
> [1] Qinghua Liu and Praneeth Netrapalli and Csaba Szepesvári and Chi Jin. Optimistic MLE -- A Generic Model-based Algorithm for Partially Observable Sequential Decision Making. arXiv preprint
> arXiv:2209.14997, 2022.
>
> [2] Du, Simon Shaolei, et al. "Few-Shot Learning via Learning the Representation, Provably." International Conference on Learning Representations. 2020.
>
> [3] Cheng, Y., Feng, S., Yang, J., Zhang, H., Liang, Y. (2022). Provable benefit of multitask representation learning in reinforcement learning. Advances in Neural Information Processing Systems, 35, 31741-31754.
>
> [4] Agarwal, A., Song, Y., Sun, W., Wang, K., Wang, M., Zhang, X. (2023, July). Provable benefits of representational transfer in reinforcement learning. In The Thirty Sixth Annual Conference on Learning Theory (pp. 2114-2187). PMLR.

---

> ### Author Response · Authors · 2023-11-19
> **Response to Reviewer W5uL (Part 2/2)**
>
> **Q4:** Why is the first goal of upstream learning finding near-optimal policies for all $N$ tasks on average, as opposed to, say, finding near-optimal policies for all
>  tasks, i.e. $$\\max_n \max_{\pi} (V\_{\theta_n^*, R_n}^{\pi} - V\_{\theta_n^*, R_n} )\leq \epsilon$$
>
> **A4:** In fact, it is a standard goal for near-optimal policies for all $N$ tasks on average in multitask learning in both supervised learning [2] and reinforcement learning [3,4]. Usually, an $\epsilon$-average guarantee implies an $N\epsilon$ guarantee in the maximum sense (and does  not imply  an $\epsilon$ guarantee). This arises from the **joint** learning nature of multi-task learning. In joint learning, we cannot guarantee that each task learns equally well, and it can occur that most tasks have learned sufficiently well but a few tasks still suffer from high errors.
>
> **Q5:** Why use $\\|\cdot\\|\_{\infty}^p$ as the norm? Is it even a norm (i.e., satisfies the conditions required)?
>
> **A5:** Yes, the $\ell_\infty$ norm $\\|\cdot\\|\_{\infty}^{\mathrm{p}}$ (the $^{\mathrm{p}}$ indexes a policy) is  a {\em bona fide} norm   and satisfies the triangle inequality, homogeneity, and positive definiteness. We use $\\|\cdot\\|\_{\infty}$ as the norm because its form is easy to connect to TV (total variation) distance and its properties  can be used to control the TV distance subsequently.
>
> **Q6:** In the calculation of $\eta$-bracketing number, what is the domain of the functions? Is it the $\sigma$-algebra over $(\mathcal{O}\times\mathcal{A})^H$
> which is the domain of the distributions? Consider a simpler calculation: how to calculate the $\eta$-bracketing number for $\mathbb{P}\_{\theta}: \theta\in\boldsymbol{\Theta}\_u$? Now $\\mathbb{P}\_{\theta}$ is a probability measure which is defined over some $\sigma$-algebra $\mathscr{S}$. If it is contained in some $\eta$-bracket [$\mathbb{A}, \mathbb{B}$], then we must have $\mathbb{A}(S)\leq \mathbb{P}\_{\theta}(S) \leq \mathbb{B}(S)$ for every $S\in\mathscr{S}$. But this would imply (it might require measures to be regular, I am not sure) that $\mathbb{A}= \mathbb{P}\_{\theta} = \mathbb{B}$. So the
> $\eta$-bracketing number for $\mathbb{P}\_{\theta}: \theta\in \Theta$ becomes $|\Theta|$. I am assuming that this is not what the authors had in mind. Could you please clarify the calculation? The calculations in Appendix E are assuming the observation and action spaces are finite.
>
> **A6:**  We would like to clarify that when we say $\mathbb{P}\_{\theta}$ is a **function**, then the domain is $(\mathcal{O}\times\mathcal{A})^H$. In contrast, when we say $\mathbb{P}\_{\theta}$ is a **probability distribution**, the domain is the $\sigma$-algebra over the domain $(\mathcal{O}\times\mathcal{A})^H$. For the given example, $\mathbb{A}$ and $\mathbb{B}$ are not probability measures. Therefore, we can define them to satisfy $\sum\_{x\in\Omega}\mathbb{A}(x)<1<\sum_{x\in\Omega}\mathbb{B}(x)$, where $\Omega$ is the outcome space. Consequently, it is possible to design a **finitely many**  $\mathbb{A}$ and $\mathbb{B}$ which covers the infinite set $\Theta$. This result is  formally proved in Appendix E.
>
> **Q7:** I do not understand the discussion of pairwise additive distance based multi-task planning. Why is a distance between product distributions not sufficient as opposed to what the paper uses? Also, do the authors realize that
>  $$\sum\_n\mathtt{D}\_{\mathtt{TV}}(\mathbb{P}\_{\theta\_n}, \mathbb{P}\_{\theta_n'}) = \mathtt{D}\_{\mathtt{TV}}( \mathbb{P}\_{\theta\_1}\otimes\cdots\otimes\mathbb{P}\_{\theta_N}, \mathbb{P}\_{\theta\_1'}\otimes\cdots\otimes\mathbb{P}\_{\theta_N'}).$$
> a divergence (not distance) over product distribution?
>
> **A7:**
> In the analysis, we need to ultimately control the difference between value functions. The sum of the distances between marginal distributions can serve as an upper bound for the sum of the difference between value functions. In contrast, the distance between product distributions is not sufficient to guarantee the accuracy of the individual models of each task, and hence can not upper-bound the sum of the difference between value functions.
>
> Further, the equation on the sum of total variation distance $\sum\_n\mathtt{D}\_{\mathtt{TV}}(\mathbb{P}\_{\theta_n}, \mathbb{P}\_{\theta\_n'})$ given by the reviewer does not hold in general, because the TV distance is not "additive" (unlike the KL divergence, for example). In particular the left-hand side (LHS) can be greater than the right-hand side (RHS). Notice that LHS can be greater than 1, but RHS cannot exceed 1 (because the TV distance is bounded above by 1). An example is to consider that all $\theta\_n$ are identical (to say $\theta$), and all $\theta\_n'$ are identical (to say $\theta'$) but $\theta\ne\theta'$, and the total variation distance between each pair of them is 1. Then, the LHS grows with $N$, while the RHS is at most 1.

---

> > ### Comment · Reviewer_W5uL · 2023-11-21
> > **Reply to authors**
> >
> > Thank you for the detailed response. A reread of the paper while keeping these points in mind makes the discussion much more clear. I have increased my score to reflect this.

---

> > > ### Author Response · Authors · 2023-11-21
> > >
> > > We would like to appreciate the reviewer's decision to increase the score and are grateful for your thorough review!

---

### Official Review · Reviewer_t71h · 2023-11-08

**Soundness:** 3 good
**Presentation:** 3 good
**Contribution:** 3 good
**Rating:** 6
**Confidence:** 3

**Summary:**

This paper studies the problem of multi-task reinforcement learning under non-Markovian decision making process. By assuming the multi-tasks share the same action and observation spaces, and the models are from a certain parameter class, the sample complexity of learning an averaged optimal multi-task policy (using UMT-PSR, an proposed algorithm) is given in Theorem 1. The complexity is related to the complexity of the parameter class, which is measured by $\\eta$-bracketing numbers. This result shows the benefit of multi-task learning when compared with learning tasks separately. For the downstream class learning, by adopting OMLE, the sample complexity is given in Theorem 2, which is related to the complexity of the downstream model class (constructed by upstream learning) that can be reduced by previous upstream learning. The authors also instantiate their generic framework on three examples.

**Strengths:**

Originality: Studies the combination of non-Markovian process and multi-task RL, which is a relatively unexplored topic.
Quality: provide generic frameworks together with concrete examples, which shows the applicability of this theoretical analysis.
Clarity: the writing is smooth, the ideas and intuitions are also clear.

**Weaknesses:**

The downstream learning seems to be only applying previous results on a smaller downstream model class, without further new ideas.

**Questions:**

1. can these results generalize beyond low-rank problems? (maybe not low-rank, but some other structures)
2. The results hold for finite action and observation spaces. Can they be generalized to general infinite spaces? (maybe using function approximation or other techniques)

---

> ### Author Response · Authors · 2023-11-19
> **Response to Reviewer t71h**
>
> We thank the reviewer for the careful reading and thoughtful comments. We address the reviewer's questions in the following and revised the paper accordingly. The changes are marked in blue in our revision. We hope the responses below and changes in the paper address the reviewers' concerns.
>
> **Q1:** The downstream learning seems to be only applying previous results on a smaller downstream model class, without further new ideas.
>
> **A1:** Thanks for the comments. In fact, our downstream analysis involves the following novel development that is not in previous results. Since the downstream model class is not known in advance, the agent needs to use information from the upstream to approximate the model, which introduces an approximation error $\epsilon_0$. Our handling of such an error here features a novel technique of using **Renyi divergence** to measure the approximation error, which guarantees the sub-optimality bound **without requiring the realizability condition**. Such a technique has not been developed in previous results.
>
> **Q2:** Can these results generalize beyond low-rank problems? (maybe not low-rank, but some other structures)
>
> **A2:** Thanks for the question! A potential (more general) class of problems can be general decision making problems with small **generalized Eluder coefficient** (GEC) in [1]. We expect that it is possible to extend our multi-task framework to multi-task GEC type of problems. One key step is to come up with a new complexity measure that suitably particularizes to the Eluder dimension and captures the latent structure between tasks at the same time.
>
> **Q3:** The results hold for finite action and observation spaces. Can they be generalized to general infinite spaces? (maybe using function approximation or other techniques)
>
> **A3:**
> We thank the reviewer for this question. Here are our initial thoughts. For generalization to infinite observation spaces, we can possibly leverage the techniques in [2], which  extended the finite observation setting to its continuous  counterpart. We expect that similar techniques would be applicable to this proposed extension. For generalization to infinite action spaces,
> a recent study under low-rank MDPs investigated continuous action spaces [3]. Similar to [3], with some smoothness assumptions, we expect that these techniques might be used  to generalize our results to infinite continuous action spaces.
>
> [1] Zhong, H., Xiong, W., Zheng, S., Wang, L., Wang, Z., Yang, Z., Zhang, T. (2022). Gec: A unified framework for interactive decision making in mdp, pomdp, and beyond. arXiv preprint arXiv:2211.01962.
>
> [2] Qinghua Liu and Praneeth Netrapalli and Csaba Szepesvári and Chi Jin. Optimistic MLE -- A Generic Model-based Algorithm for Partially Observable Sequential Decision Making. arXiv preprint
> arXiv:2209.14997, 2022.
>
> [3] Bennett Andrew, Nathan Kallus, and Miruna Oprescu. "Low-Rank MDPs with Continuous Action Spaces." arXiv preprint arXiv:2311.03564 (2023).

---

### Official Review · Reviewer_iJpQ · 2023-11-10

**Soundness:** 3 good
**Presentation:** 4 excellent
**Contribution:** 3 good
**Rating:** 8
**Confidence:** 2

**Summary:**

This paper investigates the (upper-bound) benefits of multitask learning in PSR environments vs per-task single-task learning. The setting is episodic RL and the proposed algorithm constructs and refines a confidence set of candidate environment parameters, and in each iteration uses those parameters to compute the data collection policy.

Besides introducing the RL algorithm for multitask learning, the paper gives performance bounds for multitask learning and transfer to downstream tasks. It also explores the bounds in specific PSR settings, compared to bounds from using separate RL methods for each task. The comparisons highlight the advantages of the multitask approach.

The key algorithmic idea is to maintain a joint confidence set for the potential per-task environments, and the key technique used in the bound is considering a covering number of this confidence set. The log of this covering number grows slower or sometimes much slower with the number of tasks, than the sum of log-covering numbers of separate, per-task confidence sets.

**Strengths:**

The paper is well written and clear. Overall I am happy with this paper. It is clearly written and easy to follow. My reading of the contributions is that they offer a better understanding of multitask PSR problems more than offering a solution to multitask learning in PSRs. This is mostly because the PSRs are assumed to be given and the algorithm uses components that are useful for a theoretical study but tricky to set up in practice (Though it would be great to hear from the authors with details if they disagree.) To me the main takeaway is that, through studying upper-bounds, we can say that multitask PSRs are easier to learn (jointly), with interesting examples of shared structures across tasks that accelerate learning.

**Weaknesses:**

I do not see any issues with the paper.

**Questions:**

There is another family of PSRs that I would like to suggest as an example (from personal experience) and they can be relevant for the current and future work:
* A set of tasks that is not observed uniformly. We still want uniformly good behavior on all the tasks, but the algorithm can only sample the tasks from a distribution, rather than go over each of them one by one in each iteration. This is highly relevant to how we train RL agents in POMDPs with procedural generation of the initial state, because procedural generation gives very coarse control over the resulting initial state distribution. Here, I am seeing each initial state in the support of the distribution as a task. The downstream learning is also interesting for this.
* A single PSR with block structure in the dynamics. This is like the example above, but the multiple tasks are not explicitly recognized as such.

Some things I would like to see in the downstream regime:
* What impact does more and more training on the upstream tasks have on the zero-shot performance on the downstream task?
* What impact does more and more training on the upstream tasks have on the speed of learning on the downstream task? It would be a surprising and interesting find if some amount of "pre-training" upstream would actually improve the rate of convergence of the downstream. I guess it's more likely that the guarantee would be like "if you want to train for a given budget X downstream, then you can get good rates if you train for Y amount of experience upstream."

At a higher level, not as a criticism to the paper, though, I find the overall setting a bit odd. The proposed algorithm does not have any sequential interaction with the environment. Instead it runs the policy in the tasks for collecting data and updates its confidence set. What I find odd is therefore that the tasks can be so hard intra-episode that there is nothing we can do by adapting as we act, and we might as well pick policies, deploy them, and update. I guess the setting also does not quite apply to the kinds of environments that would be "easy" and where intra-episode adaptation could help improve performance.

I liked the fact that the results allow us to recover the batch setting when we train on N copies of the same task.

I am somewhat confused about what $\pi(\omega_h)$ means in Eq. 2, considering that $\pi$ for the end of the episode $\omega$ depends on what happened in the beginning of the episode ($\tau$). So how does $\tau$ factor into Eq. 2?

It would also be nice to understand the relationship between r from Definition 1 and multiple tasks. Considering the correspondence between block-dynamics and multiple tasks I mentioned above (that is, multiple tasks can be put together into a single POMDP that samples the task as part of the initial state), what is the relationship between r and N? r is arguably harder to scrutinize than N and the shared structure between tasks, so maybe it's possible to get rid of r as a proxy for multiple tasks and formalize everything in multitask terms?

Typos:
* I was a bit confused reading algorithm one because it refers to quantities that are defined after it is shown ($\nu^{\pi_n,k}$).
* In Example 3, the P for the core tasks seem to be on the wrong font type?

---

> ### Author Response · Authors · 2023-11-19
> **Response to Reviewer iJpQ (Part 1/2)**
>
> We thank the reviewer for the careful reading and thoughtful comments. We address the reviewer's questions in the following and revised the paper accordingly. The changes are marked in blue in our revision. We hope the responses below and changes in the paper address the reviewers' concerns.
>
>
>
>
> **Q1:** There is another family of PSRs that I would like to suggest as an example (from personal experience) and they can be relevant for the current and future work:
>
> A set of tasks that is not observed uniformly. We still want uniformly good behavior on all the tasks, but the algorithm can only sample the tasks from a distribution, rather than go over each of them one by one in each iteration. This is highly relevant to how we train RL agents in POMDPs with procedural generation of the initial state, because procedural generation gives very coarse control over the resulting initial state distribution. Here, I am seeing each initial state in the support of the distribution as a task. The downstream learning is also interesting for this.
>
> **A1:** Thanks for the great questions and suggestions! Concerning the case that tasks are not observed uniformly, our current framework should be able handle this example with a slight change of the algorithm, namely, that  the confidence set should be modified so that it is  constructed using only the data observed. The data-collection step (Line 4) should be changed from ``collecting data for all task n`` to ``only collecting data for certain task n sampled from the prior distribution`` $\mathcal{D}_N$. Then, we only use the collected data to construct the confidence set (Line 7). In this way, we surmise that the learning goal should be modified to $\mathbb{E}\_{n}[V\_{\theta\_n^*, R\_n}^{\pi^*} - V\_{\theta\_n^*, R\_n}^{\pi\_n} ] \leq \epsilon$, where task $n$ follows the distribution related to $\mathcal{D}\_N$.
>
> **Q2:** A single PSR with block structure in the dynamics. This is like the example above, but the multiple tasks are not explicitly recognized as such.
>
> **A2:** Thanks for the example, which does not explicitly contain multiple tasks but the single PSR has additional structure that a suitably designed algorithm can possibly exploit. We envision that this setting will require more careful thinking and analyses because the algorithm first has to **identify** that the PSR indeed contains the additional structure. Even if this is known, the blocks must be uncovered for further exploitation. A possible solution is to design an additional mechanism which estimates the relationship between the current observation and its underlying environment. We envision that the nature of the result would be rather different. In particular, a quantity that is distinctly different from our $\eta$-bracketing number would emerge as a quantification of the utility of the block structure in the single PSR.
>
> **Q3:** What impact does more and more training on the upstream tasks have on the zero-shot performance on the downstream task?
>
> **A3:** In the zero-shot case, since there is no interaction with the environment for the  downstream tasks, the performance will be  determined solely by the approximation error $\epsilon\_0$ of using upstream information to estimate shared structure with downstream model. To wit, the more the training data utilized by the upstream tasks, the smaller the $\epsilon\_0$, and the better the zero-shot performance. Since the upstream task may not fully capture all model parameters of the downstream task, in practice, if possible, few-shot interaction with the downstream environment is typically   recommended to improve the overall performance.
>
> **Q4:** What impact does more and more training on the upstream tasks have on the speed of learning on the downstream task? It would be a surprising and interesting find if some amount of "pre-training" upstream would actually improve the rate of convergence of the downstream. I guess it's more likely that the guarantee would be like "if you want to train for a given budget X downstream, then you can get good rates if you train for Y amount of experience upstream."
>
> **A4:** In some sense, our Theorem 2, which characterizes the benefits of downstream transfer learning provides such type of guarantee. Theorem 2 provides the total variation distance between estimated model and true model, which is a function of the log bracketing number $\sim \beta\_0$ and the approximation error from upstream learning $\epsilon\_0$. Specifically, the rate of learning (convergence speed) is
> inversely proportional to $\beta\_0$, which scales with $\epsilon\_0$. Similar to our answer to the above question, the more the training data of upstream tasks, the smaller the approximation error, and the faster the speed of learning.

---

> ### Author Response · Authors · 2023-11-19
> **Response to Reviewer iJpQ (Part 2/2)**
>
> **Q5:** At a higher level, not as a criticism to the paper, though, I find the overall setting a bit odd. The proposed algorithm does not have any sequential interaction with the environment. Instead it runs the policy in the tasks for collecting data and updates its confidence set. What I find odd is therefore that the tasks can be so hard intra-episode that there is nothing we can do by adapting as we act, and we might as well pick policies, deploy them, and update. I guess the setting also does not quite apply to the kinds of environments that would be "easy" and where intra-episode adaptation could help improve performance.
>
> **A5:** Thanks for the comment! The reason that there is no intra-episode adaptation is that in the setting we consider which involves sequential decision making with a finite horizon, the transition kernel within an episode is **heterogeneous**. Hence, the data in the previous time steps with one episode does not provide any useful information in the subsequent time steps. As a result, there is no benefit  in  updating the policy for later steps using the earlier information. Instead, the policy update is made after each entire episode which does provide useful information for later episodes. We would like to further point out that in some ``easy`` case where the transitions share some similarities inside one episode, adapting the policy within the episode can improve performance.
>
> **Q6:** I am somewhat confused about what $\pi(\omega_h)$ means in Eq. 2, considering that $\pi$ for the end of the episode $\omega$ depends on what happened in the beginning of the episode ($\tau$). So how does $\tau$ factor into Eq. 2?
>
> **A6:** We apologize for the confusion and the error in Eq. (2) which should also be maximized over all possible $\tau_h$. Eq. (2) should be
> $$
>     \forall h \in [H], \max\_{x\in\mathbb{R}^{d\_h}:\|x\|_1\leq 1} \max\_{\pi\in\Pi}\max\_{\tau_h \in \mathcal{H}\_h} \sum\_{\omega\_h \in \Omega\_h}\pi(\omega\_h|\tau\_h)|\mathbf{m}(\omega\_h)^{\top}x|\leq \frac{1}{\gamma},
> $$
> where $\pi(\omega\_h|\tau\_h)=\frac{\pi(\omega\_h,\tau\_h)}{\pi(\tau\_h)}=\pi(a\_H|o\_H,a\_{H-1},\ldots,o\_{h+1},\tau\_h)\ldots\pi(a\_{h+1}|o\_{h+1},\tau\_h)$. We have clarified the definition in the revision version of the manuscript.
>
> **Q7:** It would also be nice to understand the relationship between r from Definition 1 and multiple tasks. Considering the correspondence between block-dynamics and multiple tasks I mentioned above (that is, multiple tasks can be put together into a single POMDP that samples the task as part of the initial state), what is the relationship between r and N? r is arguably harder to scrutinize than N and the shared structure between tasks, so maybe it's possible to get rid of r as a proxy for multiple tasks and formalize everything in multitask terms?
>
> **A7:** The rank $r$ measures, roughly speaking, the amount of correlation between different observations. A large $r$ indicates that there is less correlation between trajectories. However, $N$ is simply the number of potential tasks. If we combine multiple tasks into a single PSR, then $r$ measures how much information for a new task we can obtain from previously observed tasks. Therefore, $N$ is the number of tasks, while $r$ is a latent complexity measure that quantifies   the amount of correlations between tasks, similar to the $\eta$-bracketing number. The parameter $r$ should not be gotten rid of as it is a critical parameter of interest in our work as it quantifies the correlation among tasks.
>
> **Q8:** Typos: I was a bit confused reading algorithm one because it refers to quantities that are defined after it is shown ($\nu^{\pi_{n,k}}$
> ).
> In Example 3, the P for the core tasks seem to be on the wrong font type?
>
> **A8:** Thanks for the suggestion. We will adjust the algorithm so that it appears after the description. This font represents the whole environment. In terms of the "linear span", we agree that it would be better to use the notation of model dynamics $\mathbb{P}\_{\theta_1},\ldots,\mathbb{P}\_{\theta_m}$.

---

### Meta-Review · Area_Chair_LB9h · 2023-12-14

**Metareview:**

This paper considers multi-task RL under POMDPs and PSRs, and proposes a provably efficient algorithm UMT-PSR on multi-task learning and adopt OMLE for learning downstream tasks. The authors also instantiate their generic framework on three examples. This paper studies a new and important problem, overall theory is solid and sound without major weakness. The reviewers' concerns are well addressed during rebuttal, we thus recommend acceptance.

**Justification For Why Not Higher Score:**

The technics seem to be combination of existing technics of multi-task learning and learning POMDPs/PSRs. The results are not surprising.

**Justification For Why Not Lower Score:**

The overall result is sound, solid and well-written. The paper is addressing an unsolved problem.

---

### Decision · Program_Chairs · 2024-01-16

Accept (poster)